# The human side of geoscientists: comparing geoscientists' and non-geoscientists' cognitive and affective responses to geology

Anthea Lacchia*1, Geertje Schuitema1,2, Fergus McAuliffe, 1

1 University College Dublin, iCRAG, Irish Centre for Research in Applied Geosciences, UCD School of Earth Sciences, Dublin, Ireland

2 UCD (University College Dublin) School of Business, Carysfort Avenue, Blackrock, Co. Dublin, Ireland

*e-mail: anthea.lacchia@icrag-centre.org

# Abstract

Geoscientists and non-geoscientists often struggle to communicate with each other. In this paper we aim to understand how geoscientists and non-geoscientists perceive geological concepts and activities, that is, how they think (cognitive responses) and feel (affective responses) about them. To this effect, using a mixed-methods approach, we compare mental models – people's representation of a phenomenon - of the subsurface, mining/quarrying, and drilling, between geoscientists (n=24) and non-geoscientists (n=38) recruited in Ireland. We identify four dominant themes which underlie their mental models: (1) degree of knowledge and familiarity, (2) presence of humans, (3) affective beliefs, and (4) beliefs about perceived impact of the activities. While the mental models of the non-geoscientists focused more on the perceived negative environmental and economic impacts of geoscience, as well as providing evidence of lay

expertise, those of the geoscientists focused more on human interactions. We argue that
mental models of geoscientists and non-geoscientists are the result of beliefs, including
both cognitive and affective components, and that both components need to be
acknowledged for effective dialogue between the two groups to take place.

## Introduction

Geoscience activities such as mining, quarrying, hazard risk management and landscape
management are an integral part of society, affecting local communities, citizens and
scientists. In their work, geoscientists must engage and work with people from other
backgrounds and disciplines (Barthel & Seidi, 2017), as their work often directly
involves and impacts different publics (e.g. Juang *et al.*, 2019). However, geoscientists
often struggle to communicate with non-geoscientists, particularly around controversial
topics such as resource extraction and risk communication. For instance, past studies
have investigated public perception and risk communication in the case of fracking (e.g.
Boudet *et al.*, 2014; Thomas *et al.*, 2017), carbon capture and storage (Seigo *et al.*, 2014)
and earthquakes (e.g. Marincioni *et al.*, 2012). Specifically, in the context of earthquake
risk communication, Marincioni *et al.* (2012) studied the case of the 2009 earthquake in
l'Aquila, Italy, as a result of which 308 people died: the authors identified a lack of clear
communication from the risk management authorities to the public in relation to
earthquake prediction and structural resistance of buildings. In the context of public
perception of carbon capture and storage, Seigo *et al.* (2014) compared risk and benefit
perceptions of the technology in different Canadian regions, and found that predictors of
risk perceptions, such as sustainability concerns, did not vary across different regions
and were unrelated to familiarity with the technology. The authors also point out that
there is a need to address lay people's "misconceptions" related to carbon capture and
storage, in order for informed decisions to take place. In the context of a public
perceptions of fracking, Thomas *et al.*, 2017, in a literature review, identified mixed
levels of awareness of shale operations, as well as ethical issues and widespread distrust
of responsible parties. Other studies concerning fracking, such as that by Boudet *et al.*
(2014), which looked at public perceptions of fracking in the U.S., found differences in
perception between different genders, socioeconomic backgrounds, income levels and
level of education, and highlighted a need for "wide ranging and inclusive public
dialogue" around the risks and benefits of fracking. For effective, dialogic
communication (e.g. Davies and Horst, 2016; Wildson and Willis, 2004) between
geoscientists and non-geoscientists to take place, both groups must understand one
another, i.e., the audience they are engaging with (Pidgeon and Fischoff, 2011).
A starting point from which to understand each other is to investigate the differences
between geoscientists, defined as anyone with at least a university degree in geology or
geoscience, and non-geoscientists, defined as those without such a degree. While
acknowledging that those without a degree in geoscience may well possess expert
knowledge relating to geoscience, we choose to adopt these definitions as indicators of
expertise, and as useful starting points from which to discuss differences and
similarities. Specifically, we investigate these differences by adopting the concept of
mental models, which are defined for our purposes as an individual's internal
representation of a phenomenon, or a way for people to interpret and navigate the
world (Johnson-Laird, 1983, 2010, 2013; Libarkin *et al.,* 2003).
In the context of science education, Libarkin *et al.* (2003) recognise four categories of
cognitive (mental) models: "conceptual models" which are precise, highly-stable
representations of the world used by geoscientists (for instance, aquifer models);
"conceptual frameworks", organised and stable models of the world used by
geoscientists (for instance, the notion of gravity); "naïve mental models", intuitive
models of the world that so-called 'novices' fill with fragmented and unconnected
knowledge (for instance, the notion that the Earth is flat); and "unstable mental models",
unstable, incomplete and inexact mental models which are used by novices and easily
modified (for instance, the idea that the Earth is spherical, but with flattened portions
where humans live). "Conceptual mental models" are the result of cognitive change,
often due to repeated cognitive engagement with the same problems and phenomena,
and thus we envisaged that geoscientists' mental models should conform to these, and
non-geoscientists' mental models should conform to Libarkin's "naïve mental models"
or "unstable mental models", as they are typically based on intuition and local
knowledge.
Mental models have previously been used to understand non-experts' perceptions of
geoscience-related topics. For instance, Bostrom *et al.* (1994) investigated non-experts'
mental models of climate change, and found that global warming was regarded as "both
bad and highly likely". Zaunbrecher *et al.*, (2018), investigating non-experts' mental
models of geothermal energy, identified varying attitudes and knowledge levels among
participants, with negative emotions being evoked by the concepts of drilling and power
stations. These studies also stress that there are emotional or affective components
underlying the mental models of non-experts.
However, most mental models studies focus merely on cognitive components (e.g.
Gibson *et al.,* 2016; Goel, 2007; Johnson-Laird, 2010, 2013; Shipton *et al.,* 2019) or on
the cognitive superiority of geoscientists over non-geoscientists (Libarkin *et al.,* 2003;
Vosniadou and Brewer, 1992). Here, we argue that mental models should also
incorporate subjective and affective representations of a phenomenon, for both
geoscientist and non-geoscientists.
Affect is a general positive or negative feeling that people may experience about an
event, a situation, a technology or a process (Finucane *et al.*, 2000). An affective
response is thus the response to such an event, situation, technology or process, based
on positive or negative feelings. Misperceptions of geological activities among the public
are often attributed to affective and emotional processes (Devine-Wright, 2005;
Finucane et al., 2000; Loewenstein et al., 2001). The role of emotions in risk perception
and communication around nuclear waste has been investigated by Sjöberg (2007), who
argued that emotions such as interest play an important role in risk perception and
attitude. In Zaunbrecher *et al.'s* (2018) study of public perception of geothermal energy,
an association between positive emotions and the acceptance of geothermal energy was
identified. Similarly, Thomas *et al*. (2015) identified negative emotions in the mental
models of non-experts when considering sea level change. While these studies recognise
emotions as a component of the mental models of non-geoscientists, far less is known
about the affective responses of geoscientists, and how they influence their mental
models, as well as how they compare with those of non-geoscientists.
Compared with the number of studies focusing on non-experts or publics, fewer studies
have used mental models to compare experts' and non-experts' perceptions. For
example, Gibson *et al*. (2016) identified mismatches in perceptions of subsurface
hydrology and geohazards between experts and non-experts. In a study comparing
experts' and non-experts' mental models of nuclear waste, Skarlatidou *et al.* (2012)
described non-experts' negative perceptions of nuclear waste as co-existing with a
positive attitude towards nuclear energy, as well as lack of knowledge and familiarity,
and discussed implications for risk communication. In the context of sea-level change,
Thomas *et al.* (2015) identified both consistencies between the mental models of
experts and non-experts, and barriers to publics engaging with the issue, and argued
that factors other than knowledge bear an influence on the mental models of non-
experts. These factors include "levels of concern, perceptions of self-efficacy and
responsibility, trust and ways of actively engaging with or avoiding the issue" (Thomas
*et al*., 2015, p.78).
The main goal of the present paper is to investigate how evaluation of both cognitive
and affective beliefs underlie the mental models of geoscientists and non-geoscientists.
We define beliefs as "psychologically-held understandings, premises or propositions
about the world that are felt to be true" (Richardson, 1996, p. 103).
To this end, we used a mixed-method approach and identified the cognitive and affective
underlying beliefs of geoscientists' and non-geoscientists' mental models. We chose to
recruit participants from a rural community in Ireland where geologists typically
conduct fieldwork (Martinsen *et al.*, 2017) because the area's spectacular Carboniferous
geology lends itself to public engagement events. Better understanding the community
will allow geoscientists and public engagement practitioners to develop such public
engagement activities. While our sample of geoscientists (n=24) working across Ireland
and non-geoscientists (n=38) recruited in a rural community in Ireland is not
representative of all geoscientists and non-geoscientists in all settings,  we suggest that
understanding differences and resemblances of both the cognitive and affective
components of mental models of geoscientists and non-geoscientists can help to
improve two-way communication between them about often-contested areas of the
geosciences.

# Materials and methods
The aim of this paper was to investigate the beliefs underlying the mental models of
Irish geoscientists vs non-geoscientists around geological concepts and activities and
use this to build future communication strategies.
To that end,  a face-to-face survey was conducted with geoscientists (n=24, recruited
across Ireland) and non-geoscientists (n= 38, recruited in a rural community in Ireland)
to compare their mental models and underlying beliefs about the subsurface of the
Earth, applied-geoscience activities (mining/quarrying and drilling), and geohazards
(flooding). To establish their mental models, respondents were asked to sketch the
activities, geohazard, and the subsurface to any depth they wished. Follow up questions
about respondents' emotions and perceived outcomes of the activities and hazard were
also included in a short survey.
In our analyses, we used a mixed experimental set-up of between-subjects design (to
compare geoscientists vs non-geoscientists) and within-subjects design (to investigate
sketches of subsurface, drilling, mining/quarrying, flooding within our sample group of
geoscientists or non-geoscientists). Moreover, a mixed methods approach was used (i.e.,
a mixture of qualitative and quantitative methods) to investigate their beliefs about the
subsurface and geological activities. Analyses of the qualitative results were done
through qualitative thematic analysis (Boyatzis, 1998; Marshall and Rossman, 1999)
and quantitative data were tested on statistical significance using the IBM SPSS Statistics
24 software package.

## Procedure
Face-to-face surveys were conducted among 24 geoscientist and 38 non-geoscientist
participants as detailed below. A summary of the socio-demographics of both is
presented in Table 1. The geoscientists who took part in the study ranged in age from 21
to 59, with most identifying as male (58%), aged 21-29, and educated to degree level.
The higher number of males is consistent with underrepresentation of females in
geoscience (Dutt *et al.*, 2016). Most non-geoscientists identified as female (63%), aged
or older and educated to less than degree level and their age ranged from 16 to 60 or
over. For a discussion of the limitations associated with our sample, see Limitations.

**Table 1. Sociodemographic details across all study participants.**

| | Geoscientists (n) | Non-geoscientists (n) |
|---|---|---|
| **Female/ Male** | 42% females/ 58% males | 63% females/37% males |
| **Age** | | |
| 16-21 | 0 | 1 |
| 21-29 | 14 | 7 |
| 30-39 | 8 | 3 |
| 40-49 | 1 | 8 |
| 50-59 | 1 | 5 |
| 60 or older | 0 | 13 |
| **Declined to answer** | 0 | 1 |
| **Educational level** | | |
| less than degree level | 0 | 18 |
| to degree level | 14 | 16 |
| Other (higher than degree level) | 10 | 4 |



Non-geoscientists were recruited on several locations in County Clare, western Ireland,
between August 2017 and February 2018 (see Table 1 for socio-demographic details).
County Clare was chosen because it is a popular destination for geoscientists from
academia and industry in the Republic of Ireland (e.g. see Martinsen *et al.,* 2017). It is an
excellent setting for non-geologists to learn about geology, as well as one of the top
tourist destinations in Ireland. Given the popularity of the area with geologists, we also
anticipated that non-geoscientists living in the area may have a relatively high level of
familiarity with geology or with groups of geologists, thus potentially providing useful
insights for dialogue in this community.
Invitation letters were posted to 50 addresses selected randomly using the online (Eir)
phonebook and follow-up telephone calls were made to schedule a time for the survey
to take place. In the invitation letters, participants were asked to take part in a study
investigating public perception of geology, including knowledge about the geology of Co.
Clare and the subsurface. No specific information on the aims of our study was provided
in order to minimise response bias. This method was supplemented by convenience
sampling in local businesses in Co. Clare. Details of those who did not wish to participate
were immediately destroyed. Before commencing any interviews, following University
College Dublin's ethical guidelines, all interviewees provided informed consent.
No incentives were offered for participation. The survey was administered in person by
the lead author. Each survey took approximately 20-30 min to complete. Relevant
spoken quotes by respondents during survey completion were written down by the lead
author as support information and were included in the analysis.
Geoscientists were defined as people with a degree in geoscience, either working or
doing research in the geosciences. They were recruited using convenience sampling
techniques and ranged from MSc students (n=1), PhD students (n=11), and postdoctoral
researchers (n=7), to professional geoscientists working in geoscience industry and
academia (n=4) or education centres (n=1).
All participants were offered the opportunity to have the results of the research sent to
them by sharing their contact details. Contact details were immediately separated from
the data to guarantee anonymity.

## Face-to-face survey

The survey was aimed at qualitatively assessing underlying beliefs of respondents'
mental models of the subsurface, drilling, mining/quarrying, and flooding. This
qualitative analysis was supplemented by quantitative analysis of survey responses.
First, respondents were asked: 'please sketch the ground under your feet starting from
the surface of the earth down to any depth'. They were then asked to make sketches of
drilling, mining/quarrying and flooding, a common way of measuring mental models
(e.g. Gibson *et al.*, 2016).
For drilling, mining/quarrying, and flooding, there were follow-up quantitative
questions on the environmental and economic impacts, and the emotions associated
with the activities and hazard. Flooding did not yield reliable scales for affective
responses or significant results for perceived impact, hence it was excluded from further
analyses and from the rest of the results.
Perceived environmental and economic impact of the activities were measured on a 5-
point Likert scales ranging from totally disagree (1) to totally agree (5). To measure the
perceived economic impact, after each sketch (of drilling and mining/quarrying)
respondents were asked whether drilling or mining/quarrying *will improve the local*
*economy*. Perceived environmental impact was measured by asking whether drilling or
mining/quarrying *will have a negative impact on the local natural environment.*
Next, respondents were asked to rate how well a given emotion described their feelings
towards drilling and mining/quarrying, respectively. They indicated which feeling they
identified with from a list of 16 different feelings on 5-point bipolar scales, of which 8
were negative emotions (i.e., irritated, angry, hostile, frightened, frustrated, upset,
concerned, deceived) and 8 positive emotions (i.e., optimistic, satisfied, inspired,
enthusiastic, relaxed, excited, safe and interested). The measures were based on scales
previously used by Sjöberg (2007), Roderiquez *et al.*, (2018), and Visschers and Siegrist
(2014). The negative and positive affective responses both formed reliable scales (Table
2), which is indicated by scores of Cronbach's alpha of 0.70 or higher (Peterson, 1994),
and the mean scores on negative and positive affective responses were computed and
used in further analysis.
**Table 2. Reliability, mean (M) and standard Deviations (SD) of scales of affective**
**responses and perceived impact.**

| | Geoscientists | | | Non-geoscientists | | |
|---|---|---|---|---|---|---|
| | Cronbach's Alpha | M | SD | Cronbach's Alpha | M | SD |
| **Affective responses** | | | | | | |

| | | | | | | |
|---|---|---|---|---|---|---|
| Negative affect drilling | 0.881 | 1.49 | 0.61 | 0.918 | 2.32 | 1.02 |
| Positive affect drilling | 0.944 | 3.19 | 1.12 | 0.953 | 2.40 | 1.09 |
| Negative affect mining/quarrying | 0.853 | 1.42 | 0.53 | 0.886 | 2.28 | 0.97 |
| Positive affect mining/quarrying | 0.958 | 3.02 | 1.22 | 0.835 | 2.22 | 0.87 |
| **Perceived impact** | | | | | | |
| Economic impact drilling | N/A | 3.40 | 1.27 | N/A | 2.62 | 1.08 |
| Economic impact mining/quarrying | N/A | 4.05 | 1.39 | N/A | 2.94 | 1.35 |
| Environmental impact drilling | N/A | 2.16 | 0.92 | N/A | 3.48 | 1.39 |
| Environmental impact mining/quarrying | N/A | 3.05 | 0.80 | N/A | 3.74 | 1.22 |

Note: Whenever Cronbach's Alpha was not relevant (i.e., for single items) N/A is written in the table.

# Analysis strategy

## Analysis of the sketches

The first and second authors examined the sketches using a grounded theory approach

taken as "the progressive identification and integration of categories of meaning from

data" (Willig, 2008, p.35). This allowed the identification of six indicators of knowledge and familiarity in the sketches, namely: amount of *technical jargon*, defined as the presence of technical and subject-specific vocabulary in the labels of sketches, *sense of scale,* which refers to an indication of the awareness of the size of different elements included in the sketches (usually provided by a point of reference such as a scale bar); *number of layers,* the number of layers of rock or other material in the sketches; *number of labels,* the number of labels included in the sketches; *depth*, which refers to the depth to which they sketched the subsurface, ranging from the ground surface (coded as 1) to the core (5); and *human interactions*. The authors scored the sketches independently based on this. Pearson's correlation was used to determine the inter-rater reliability, which was deemed acceptable (Pearson's $r \geq 0.7$, $p \leq 0.001$).

To test the differences between geoscientists and non-geoscientists on the six pre-defined indicators, Independent Sample T-tests and ANOVA Repeated Measures analyses were conducted using the IBM SPSS Statistics 24 software package.

These results informed our qualitative analysis of the sketches, whereby the sketches were subsequently analysed by means of thematic analysis to identify themes that were common to some or all of the sketches (Boyatzis, 1998; Marshall and Rossman, 1999). Thematic analyses were conducted manually by the first author.

## Analyses of perceived impact and affective responses

As we had a mixed design of between-subjects (geoscientists vs non-geoscientists) and within-subjects (drilling and mining/quarrying), we conducted two ANOVA Repeated Measures with geoscientists and non-geoscientists as between-subjects variables and perceived impact and affective response as dependent variables, respectively. Posthoc t-

tests as part of the ANOVA Repeated Measures were run to compare in detail the
cognitive and affective responses of geoscientists and non-geoscientists.

# Results

Thematic analysis was used to analyse all sketches and written comments on the survey.
We identified four common themes: (1) knowledge and expertise relative to the topics,
(2) beliefs about human interactions (presence of humans in the sketches), (3) affective
beliefs, and (4) beliefs about the impact on the economy or environment.

## Knowledge and expertise

*Technical knowledge and familiarity*
The mental models of geoscientists contained indicators of detailed, technical
knowledge and familiarity with geoscience content stemming from years of training and
from professional expertise (e.g., see Cronin *et al.,* 2004). Specifically, the sketches made
by geoscientists extended down to a greater *depth*, included more *technical jargon*
related to geoscience, more *labels*, more *layers* within the Earth's interior, and a greater
*sense of scale*, compared to those of non-geoscientists (Fig. 1). For instance, it was
common for geoscientists to extend their sketches down to the mantle and/or core.
It is not surprising that geoscientists included these indicators of technical knowledge in
their sketches given that drawing and sketching the landscape and the Earth's interior
are skills typically acquired during geoscience undergraduate education (Johnson &
Reynolds, 2006) and given the importance of spatial visualisation as a geoscience skill
(Titus & Horsman, 2009). Without being prompted to do so, some geoscientists also
included colours and colour-coding in their sketches, which is another habit likely to
have been acquired during undergraduate geoscience training and thus linked to
technical knowledge. Geoscientists may also have enjoyed the task of sketching to a
greater extent, wanting to provide as much information as possible: for instance, a sense
of enjoyment was reflected in the inclusion of smiles on the faces of stick figures in one
geoscientist's sketch, which also included different types of fossils and crystal shapes
(Fig. 1g). It was not uncommon for geoscientists to include exclamation marks in their
labels, such as "*Hawaii!*", indicating engagement with the process of sketching and
enjoyment. A greater degree of technical knowledge and familiarity with geoscience in
the sketches of geoscientists is consistent with the assumption that geoscientists have
"conceptual mental models", which are developed based on their expertise and training
in geoscience.
Conversely, the lower levels of detail and technical knowledge in the sketches of non-
geoscientists may reflect lack of knowledge but may also be linked to a lack of interest in
the topics or a perception of science as inaccessible and exclusive. The notion that
science can be viewed as a distant and inaccessible entity by non-scientists was
identified in previous studies of public perception of risks (Bickerstaff *et al.*, 2006;
Michael, 1992).
Furthermore, geoscientists' comments and sketches sometimes included knowledge
that went beyond technical geoscience-related concepts, and incorporated elements of
philosophy of science. For instance, one geoscientist labelled the different layers of the
subsurface from an anthropocentric point of view as "*what we know*" (upper crust),
"*what we think we know*" (lower crust), "*where we can make an educated guess*"
(mantle), and "*anything goes*" (core). This indicates that geoscientists do not limit
themselves to technical knowledge, but also tap into other types of knowledge in
constructing their mental models. Religious belief systems also surfaced among
participants, with one non-geoscientist stating: "[...] *we disagree on that [that ammonoid*
*fossils are much older than humans]. I believe in the genesis and that humans arrived at*
*the same time as animals.*" In this case, these beliefs were deemed by the participant to
be in opposition to the science and specifically to the geoscience concept of geological
time which the survey brought to the fore.

*Lay expertise*
The non-geoscientists' sketches contained indicators of local knowledge about their own
area (Fig. 1b), which we interpret as lay expertise (e.g., Cronin *et al.* 2004; Wynne,
1996). Lay expertise is here taken as a form of knowledge that is relevant to and can
contribute to the scientific discourse (see Collins and Evans, 2002). For example, one
non-geoscientist's sketch (Fig. 1h) of mining/quarrying included historical details, such
as the historical ownership of mines by "*Judge Comyn*" and the "*government*", as well as
the location of historical phosphate mines and the past site of "*surface mining and*
*blasting*". Another non-geoscientist noted the presence of a "*water reservoir on top of*
*Black Head*" in a comment written on the sketch, while also adding at the end of the
survey: "*Having lived in Meath for 20 years, I was aware of mining in Tara Mines and the*
*creation of Newgrange Visitor Centre.*" In addition, a non-geoscientist included the
subsurface depth beneath which water could be found in their local area, alongside the
label: "*Drilling for water around Kilkee area. Good supply found*".
Such lay knowledge co-occurred with indications of low levels of familiarity and
technical knowledge relating to geological concepts and activities. For instance, when
asked to sketch the ground under their feet, one non-geoscientist included thickness of
layers at millimetre scale and labelled the layers using specific terms such as
"*ceramictite*" and "*concrete*" - indicating local knowledge - but did not know what was
below the layer labelled "*stone, rock, clay 2m*", as is evinced from the "*? ? ? ?*" label (Fig.
1b), indicating uncertainty or unfamiliarity. Uncertainty was similarly expressed
through written notes accompanying the sketches such as "*not sure*", "*Cannot envisage*
*this enough to draw. Sorry.*" or "*no idea how far down that goes*". This sense of
uncertainty may also be linked to the sense of distance from science viewed as exclusive
and inaccessible already described.
*Concluding remarks*
In conclusion, even though the mental models of non-geoscientists contain few
indicators of technical knowledge and familiarity, they possess lay knowledge, which is
valuable for geoscientists and is for example recognised in citizen science projects that
include the non-geoscientists in research projects (e.g., Nature, 2018; Skarlatidou *et al.,*
2012; Vera, 2018).
Therefore, while at first glance it appears that geoscientists possess conceptual mental
models and non-geoscientists possess naïve mental models, given that geoscientists
have more familiarity and technical knowledge related to geoscience, we find that
underlying this, the mental models of both geoscientists and non-geoscientists are
complex and reflect different knowledge in both groups.

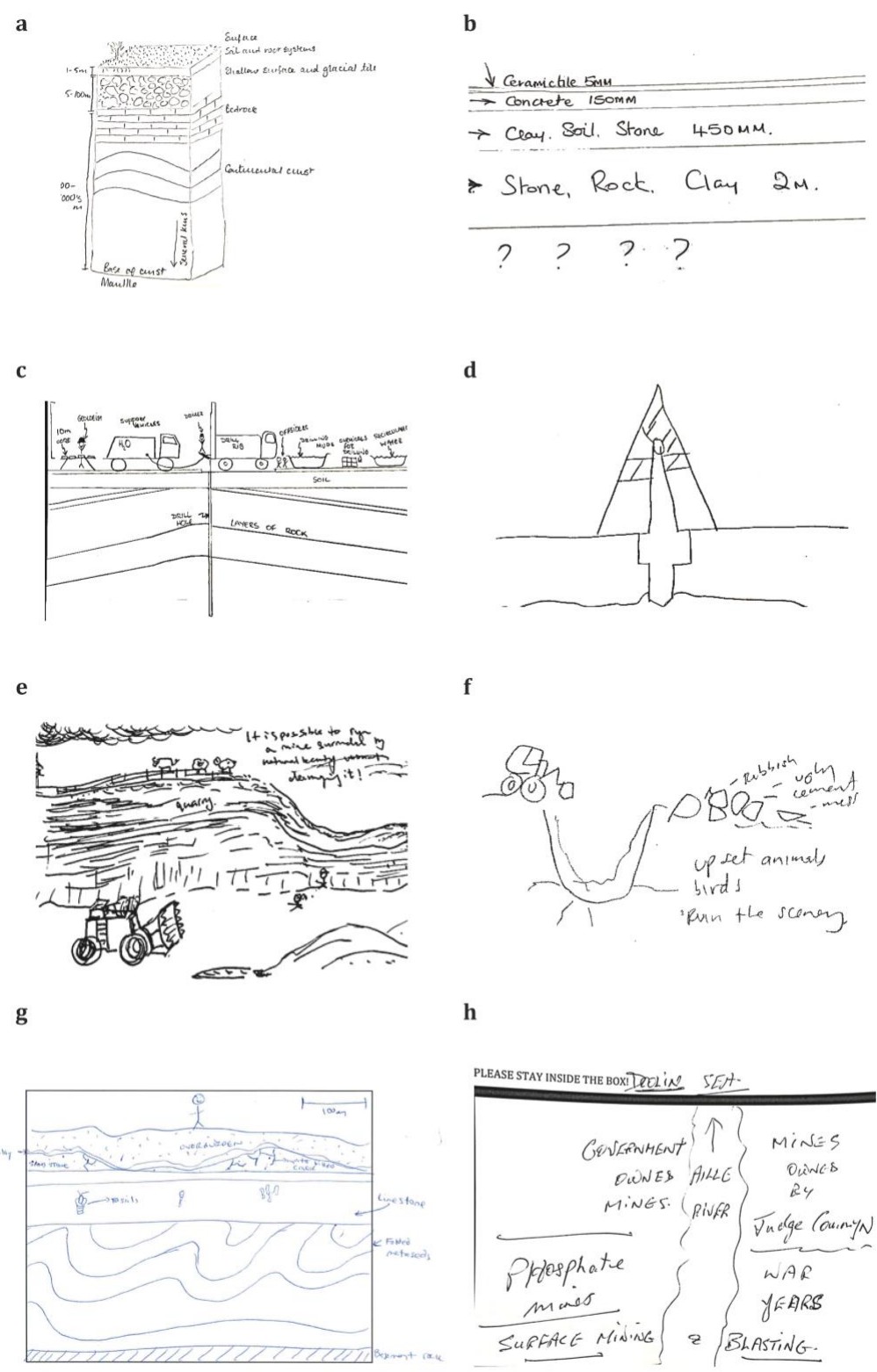


Fig. 1. Comparison of sketches made by geoscientists (left column) and non-geoscientists (right
column). The sketches are of: **a,b**, the subsurface; **c,d** drilling; **e,f**, mining/quarrying; and **g,h,**
subsurface (left), and mining/quarrying (right).

## Beliefs about human interactions

A second theme that emerged from the sketches was the number of human interactions,

defined as the presence of humans or human-operated machines in the sketches,

comments or labels, including human-built structures such as a field, road or house.

Geoscientists' sketches typically included human interactions. In particular,

mining/quarrying activities were sketched from a very human lens by geoscientists,

who highlighted details of people working in a lab or processing plant, or people using

instruments such as microscopes (Fig 1c). Geoscientists also included details of labour

division, showing people with tools performing different functions, or stick figures with

hammers or helmets doing different types of work (Fig. 1c,e).

Non-geoscientists included fewer human interactions in their sketches, but contributed

to the human interaction theme in their written comments in a different way. For

instance, one non-geoscientist wrote: *"People are not interested in geology*". These

results contrast with earlier reports of an anthropocentric view of the subsurface on the

part of non-geoscientists, with geoscientists focusing on technical geoscience concepts

rather than on human elements (e.g., Gibson *et al.,* 2016). A possible explanation is that

mining/quarrying and drilling are tied to geoscientists' jobs and therefore including

humans in the sketches may be geoscientists' way of highlighting the social process of

science and their work.

These findings on human interactions are confirmed by Independent Sample T-tests,

which indicate that geoscientists included more human interactions than non-

geoscientists when sketching drilling, $[t(56) = 3.77, p \leq 0.001]$ and mining/quarrying,

$[t(56) = 3.14, p = 0.003]$ . It is worth noting that, for the purposes of this analysis, a

group of humans close together in the sketch was counted as one human interaction.

## Affective beliefs

Drilling and mining/quarrying are highly controversial geological activities, and therefore we asked geoscientists and non-geoscientists to indicate their affective responses to them (see method), which refers to a general positive to negative feeling about these geological activities (Visschers and Siegrist, 2008). An ANOVA repeated measures analysis revealed a significant interaction effect, (Wilks' $\lambda$ = 0.76); [$F_{(3,57)}$= 5.977, $p \leq 0.001$], indicating that geoscientists and non-geoscientists have different affective responses to drilling and mining/quarrying.

As illustrated in Fig. 2, the posthoc tests effect revealed that non-geoscientists had more negative affective responses to mining/quarrying, [$t(59)$ = -3.96, $p \leq 0.001$], and drilling, [$t(60)$ = -3.69, $p \leq 0.001$], compared to geoscientists. Instead, geoscientists had more positive affective responses to mining/quarrying [$t(59)$ = 2.94, $p$ = 0.004], and drilling, [$t(60)$ =2.85, $p$ = 0.005], compared to non-geoscientists. Geoscientists had far more positive than negative affective responses to both drilling and mining/quarrying, whereas non-geoscientists' strength of positive and negative affective responses did not statistically differ.

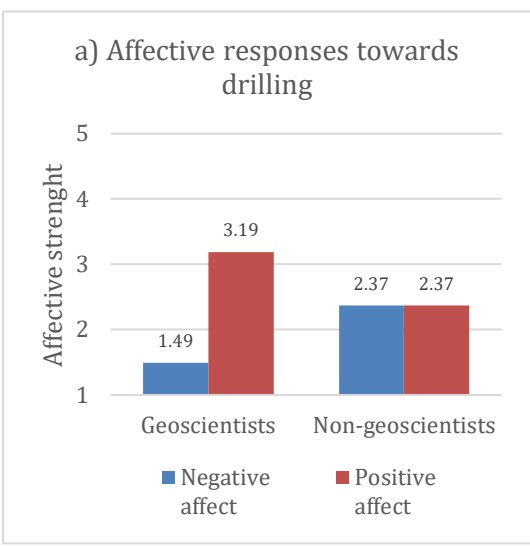 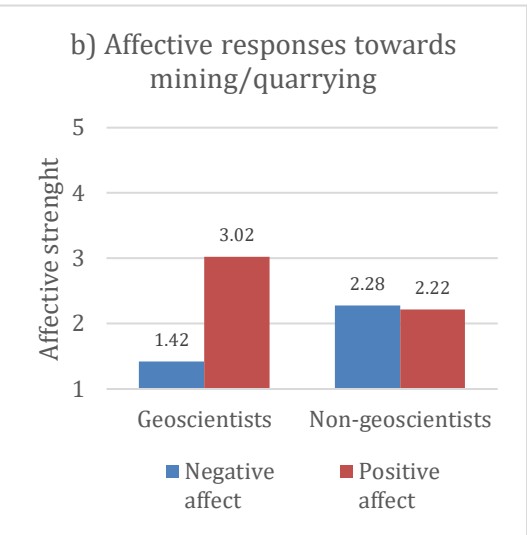

Fig. 2. a,b Affective responses towards drilling and mining/quarrying. Mean values of positive
and negative affect responses are compared between geoscientists and non-geoscientists for
different activities, namely (a) drilling and (b) mining/quarrying;  measurements are on a scale
from 1 (weak affective strength) to 5 (strong affective strength).

It should be pointed out that many of the geoscientists in our sample worked in research
in geoscience activities (though area of research was not formally gathered), which
could have resulted in more positive affective associations with their field of research,
such as feelings of safety (cf. Mearns and Flin, 1995).

## Beliefs about environmental and economic impact

An environmental or economic impact theme emerged from thematic analysis of the
sketches. Non-geoscientists' sketches often highlighted environmental effects of drilling
and mining/quarrying activities (e.g., noise from drilling, environmental degradation or
pollution) through labels (Fig 1f), indicating that negative environmental impacts were
at the forefront of their mind. For instance, this was illustrated by labels such as *"Grassy*
*bank 3-4m high to screen activity from the outside world as process is unsightly"*. The
theme was also present in written comments by non-geoscientists, such as: *"I live on the*
*River Shannon where we have a large colony of dolphins. Several years ago a company*
*wanted to open a quarry that requires blasting up to 3-6 times a week. Locals objected to*
*this blasting as we believed that the blasting would affect the dolphins by way of seismic*
*waves travelling through the ground and out to the Shannon. WE WON!"* Another non-
geoscientist, when sketching rock drilling, wrote "*causing underground problems, release*
*of gas, etc., poisoning wells etc*." In general, it was clear that the non-geoscientists tended
to associate negative emotions with the negative impact of geoscience on the
environment, such as in the label "*ruin the scenery, upset animals, birds*" (Fig. 1f).

Through their labels, non-geoscientists also reported concern about the negative effects
of geoscience on the economy (e.g., loss of tourism), as for example evinced by the label
"*Road networks e.g. quarries, need to be in the Shannon [area] – this is a tourist area, not*
*here"*. One label by a non-geoscientist is taken to imply a lack of trust in how geoscience
operates: *"I think it is unfortunate that most geological studies are funded by large*
*industry*". Lack of trust in industry and government has previously been identified as a
dominant theme in a review of public perceptions of hydraulic fracturing for shale gas
and oil (Thomas *et al.*, 2017).

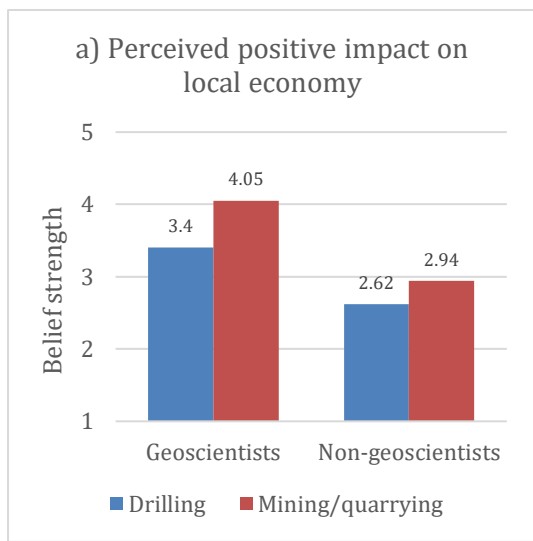 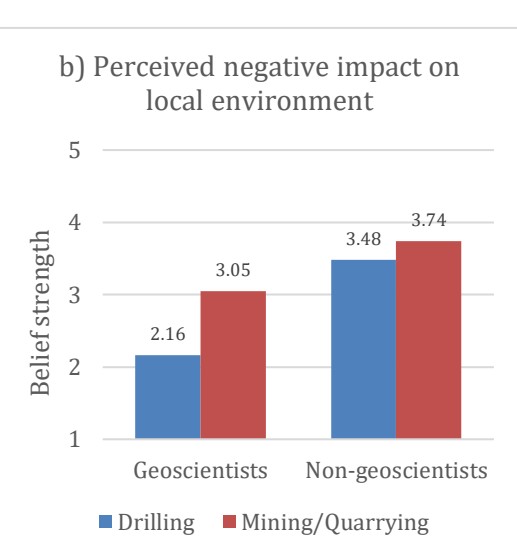


Fig. 3. Perceived economic and environmental impact. (a) Mean scores in answer to beliefs on the extent to which they agreed that drilling and mining/quarrying would improve the local economy; (b) Mean scores in answer to beliefs on the extent to which they agreed that drilling and mining/quarrying would have a negative impact on the local natural environment; measurements are on a scale from 1 (totally disagree) to 5 (totally agree).

These conclusions were confirmed in additional survey questions about the effects of drilling and mining/quarrying on the local economy and environment (see method). An ANOVA repeated measures analysis showed a significant interaction effect: geoscientists and non-geoscientists differed in their beliefs about impact across the geological activities of drilling and mining/quarrying, (Wilks' $\lambda$ = 0.773); [$F(3, 57)$ = 5.578, $p$ = 0.002]. Specifically, non-geoscientists perceived greater negative impacts on the local environment for drilling, [$t(49)$ = -3.59, $p$ = 0.02], and mining/quarrying, [$t(51)$ = -2.15, $p$ = 0.036], compared to geoscientists. In contrast, geoscientists perceived greater positive impacts on the local economy from drilling, [$t(55)$ = 2.43, $p$ = 0.019], and mining/quarrying, [$t(56)$ = 2.92, $p$ = 0.005], compared to non-geoscientists (Fig. 3).

In line with previous studies of perceptions of the underground (Partridge *et al.*, 2019), we recognised tensions between economic values and environmental values in comments written on the survey, such as "*Drilling for a well for water is ok. Drilling for oil or gas is not necessary. Invest in solar and wind energy alternatives. Fracking is just idiotic.*" Such comments tended to equate fracking with a threat, associated with fear. Another participant wrote: "*Concerned about fracking if not properly supervised*". This tension may be linked to a desire for control (cf. Hooks *et al*. 2019) and regulation of geoscience activities and technologies (e.g., GSI, 2016), as typified by comments such as "*Concerned about fracking if not properly supervised*" or "*Groundwater pollution with farming practices, I would like it to be more controlled.*"

Geoscientists, while indicating an awareness of the negative effects of geoscience on the
environment in written comments on the survey, generally downplayed the negative
effects and were sometimes defensive in tone. For example, one geoscientist while
answering that mining/quarrying would lead to an increase in numbers of visitors and
tourists to the area, wrote: "*Giving you an example, in North Yorkshire* [UK]*, there is a salt*
*mine near Staithes where tourists are attracted by its geology and natural beauty. The*
*mine is not necessarily degrading the importance of the land as a long as [there is] a good*
*system keeping it in place.*" Another label written by a geoscientist illustrates a defensive
tone: "*It is possible to run a mine surrounded by natural beauty without damaging it!*"
(Fig. 1g).
In conclusion, beliefs about the environmental or economic impact underlie the mental
models of both geoscientists and non-geoscientists, which suggests that they both are
concerned about how geoscience activities impact the environment and economy.
However, while geoscientists tended to highlight the positive impacts, often in a
defensive tone, non-geoscientists tended to dwell on the negative ones.

# Discussion

We have highlighted the differences in mental models between a sample of Irish
geoscientists and non-geoscientists and their underlying beliefs when considering
geoscience activities and concepts. We found support for our assumption that, for both
geoscientists and non-geoscientists, mental models include cognitive (based on rational
thoughts) and affective (based on feelings and emotions) components, and are therefore
not consistent with the existence of rigidly defined categories of mental models which
focus merely on cognitive components (e.g. Gibson *et al.,* 2016; Goel, 2007; Johnson-
Laird, 2010, 2013) or on the cognitive superiority of geoscientists over non-

geoscientists (Libarkin *et al.,* 2003; Vosniadou and Brewer, 1992). Indeed, we find that

the mental models of both groups are complex reflections of different knowledge, beliefs

and affect. Hence, we argue that mental models should be redefined as *the cognitive and*

*affective representation of a phenomenon*.

The presence of strong positive affective responses and human interaction in the mental

models of geoscientists contrasts with the myth of the scientist (Barthes, 1974) as an

impartial, detached observer of reality (Mitroff, 1974), and dissents with the rhetoric of

fact-based knowledge. In other words, geoscientists are first and foremost human. The

results contribute to the erosion of the ideal of the objective scientist, focused solely on

facts, helping to deconstruct the myth of science that sees scientists as impartial and

detached. Whilst the notion that all experts are affected by biases when making

judgements under uncertainty has been known by scholars at least since the work of

Tversky & Kahneman (1974), this is not commonly recognised within the geoscientific

community (e.g., see Curtis, 2012). We have shown that geoscientists and non-

geoscientists alike go beyond facts into emotional territory when constructing their

mental models.

Understanding differences and resemblances of both the cognitive and affective

components of mental models of geoscientists and non-geoscientists is an important

step in improving the communication between them, for instance when discussing

often-contested areas of the geosciences such as resource extraction (see Stewart and

Lewis, 2017). As a practical step, in communicating with each other, geoscientists and

non-geoscientists may wish to acknowledge their differences and focus on

commonalities in order to find common ground. For instance, given that both

geoscientists and non-geoscientists are concerned with the impacts of geoscience on the

economy and the environment and given that both groups incorporate affect in their

mental models of geoscience concepts and activities, geoscientists may be able to reach

wider audiences by acknowledging these concerns and affective components, and
including feelings and affect in their chosen form of communication (e.g., personal
motivations for their research). In addition, geoscientists may benefit from using
storytelling and narrative, which typically include both affective and cognitive
components, as their chosen modes of communication, a recommendation consistent
with previous science communication research (Dahlstrom, 2015). If geoscientists
acknowledge the emotional component of their mental models, this may also lead them
to reflect on the meaning of scientific knowledge and to change their view of themselves
as keepers of knowledge. On one hand, this could influence how they communicate their
work and activities to geoscientists and non-geoscientists, but it could also lead to a
broader understanding of epistemology and the social component of geoscience on the
part of geoscientists (see Stewart, 2016). While it is useful for geoscientists to
acknowledge or reflect on the affective components of their mental models, whether it is
always appropriate to incorporate emotions in communication efforts is a complex
matter that is likely to depend on the mode of communication (e.g., in person workshops
versus an explainer video on social media). There may well be occasions when the
purpose of a science communication or public engagement activity is limited to
information sharing. We suggest that, in these cases, the self-reflection brought about by
the internal acknowledgement of affective components will still be of benefit to the
geoscientists engaging in these activities.
Given that non-geoscientists often incorporate lay expertise in their mental models, in
order to build trust and common ground, geoscientists may also wish to acknowledge
and tap into local knowledge held by non-geoscientists, for example simply by asking
non-geoscientists questions about their local area. At the same time, by recognising that
geoscientists' mental models are based on emotions too, non-geoscientists may be
better able to engage with them. Overall, showcasing geoscience as a human activity
ought to help improve dialogue between the two groups. The above recommendations
are also very relevant to public engagement and science communication practitioners
who not only will be trained in how to engage with communities and publics, but are
also less likely to be seen as having an agenda (for instance motivated by economic
interests or links to industry) by the non-geoscientists they are engaging with.
*Limitations*
While this mixed-method study highlights differences and similarities between the
mental models of geoscientists and non-geoscientists, it should be noted that the sample
size is small, and thus our results need to be interpreted with care. Future research is
needed to validate our conclusions. It should further be noted that the geoscientists who
took part in this study were primarily highly-educated males working in applied
geoscience research at the time the survey took place (only 2 worked outside of
research), and they were younger compared to the non-geoscientists who took part (for
details, see Materials and Methods). The latter is fairly representative for geoscientists
(e.g., Dutt *et al.*, 2016), however, we cannot say with certainty that these differences in
socio-demographics play a role in the differences we find. For example, female and
younger geoscientists may hold different perceptions of geoscience activities and their
impacts (cf. Seigo *et al.*, 2014). However, this does not influence our main conclusion
that geoscientists' mental models are influenced by both cognitive and affective
responses.

# Concluding remarks: the human side of

# geoscientists

Our finding that geoscientists stray beyond facts into the realm of emotions and beliefs
in constructing their mental models of geoscience concepts and activities is a key
realisation for geoscience communication practitioners. We have argued that putting the
human element at the centre of communication strategies will help achieve meaningful
dialogue between geoscientists and non-geoscientists.
Geoscientists, specifically those who conduct research on resources, energy, earth and
environmental science, are increasingly required to wear multiple hats in engaging with
non-geoscientists in order to tackle societal challenges around energy and resources.
Therefore, an increased mutual understanding of the thoughts and feelings of
geoscientists and non-geoscientists will help facilitate dialogue between the two groups.

# Acknowledgements.

Ethical approval for the study was obtained from University College Dublin (Ethical
Exemption Reference Number HS-E-17-84-Lacchia-Haughton). We are very grateful to
the survey respondents who kindly gave their time to assist this research. We wish to
thank three Anonymous Referees and our editor, Beth Bartel, for their valuable
comments which much improved this manuscript.

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

## 747   Supporting information

1 Images of sketches

## Data availability

All data underlying the results is available in the manuscript and supporting information. Additional data around this project is available from the corresponding author.

## Funding

This publication has emanated from research supported in part by a research grant from Science Foundation Ireland (SFI) under Grant Number 13/RC/2092 and co-funded under the European Regional Development Fund and by PIPCO RSG and its member companies.

## Competing interests.

The authors declare no competing financial interests.

## Author contributions.

All authors conceived and planned the study; A.L. conducted the data collection; A.L. and G.S. analysed and interpreted the data; all authors helped to draft the manuscript.