# Peer review of "The human side of geoscientists: comparing"

_Geoscience Communication, 2019_

## Referee Comment (RC1) · Anonymous Referee #1 · 26 Dec 2019

The human side of geoscientists

Thank you for the opportunity to review this article, which has the potential to make a useful contribution to the field of geoscience communication. The paper is based on a sound idea and appropriate methods, but it needs work before it will be ready for publication. My main criticisms are: 1) there needs to be more critical engagement with the literature; 2) the study uses a small, biased sample; and 3) the main conclusion needs reflection. I provide more detail below.

Main comments:

- Much more engagement with the literature around perceptions of geosciences is

necessary. For example, for expert and lay perceptions of underground geology see Partridge et al (2019); Seigo et al (2014). - The conclusion that mental models are the result of beliefs that include both cognitive and affective components is not new. In two of the papers that you cite for example, the authors describe a number of 'non-knowledge' factors that contribute to risk perceptions – and you need to engage more with this literature (Sjoberg et al, 2007; Thomas et al., 2015). - The conclusion that experts are 'human' and have affective responses is also not new: see for example Wynne (1996) for a discussion of lay expertise. Some critical engagement with what constitutes expertise would also be helpful – see for example Collins and Evans (2002). - There are major biases in your sample: the geoscientists are much younger, largely students, predominantly male, and highly educated. How do you know that your results are not a function of these differences rather than the fact that they are geoscientists? A wealth of research shows that risk perceptions vary with age, gender etc – and this should be taken much more into account, as this raises serious questions for your results and conclusions. - You have some interesting qualitative findings here that deserve much more discussion. For example, what local knowledge was included? What can this tell us? What is the significance of this? Why did experts include more labels – is this anything to do with fulfilling what was expected of them? Perhaps they enjoyed it more than the lay participants so wanted to provide as much information as possible? Are geoscientists more practised in drawing diagrams, and might this explain the attention to detail? Does the amount of detail in the pictures reflect a lack of understanding or a perceived lack of understanding (the 'I'm not a scientist so I don't know' phenomenon... - see for example Bickerstaff et al 2006; Michael, 1992). Is it indifference or ignorance? There are so many things here that I would like you tell me more about. Due to the nature of your sample, I think it is difficult for you to focus on the quantitative results, but you could certainly explore your qualitative results more.

Minor comments (page/line)

- (2/29) provide some examples of why geoscience is integral in society e.g. mining,

risk management, landscape management, etc. etc. - (2/33) provide examples of problems with geoscience communication, such as with fracking and geohazards (e.g. L'Aquila earthquake). - (3/54-55) the term 'expert' would be more appropriate than 'geoscientist' as not all of this research looks at geoscientists. - (3/65) as the authors do in Thomas et al (2015, cited above). - (19/334) you mention lack of trust – you could relate this with previous research that also discusses lack of trust in geoscience industry (e.g. Thomas et al., 2017). - (21/385) – I have a couple of reservations about your conclusion that 'geoscientists are first and foremost human'. Of course they are – much research has shown that experts are human and that their judgements are based on biases etc. (see classic work by Tversky and Kahneman, for example). Furthermore, if you ask for an affective response, you will be given one - so it is no surprise that your geoscientists provided you with affective responses as well as 'cognitive' ones – it is their hobby/livelihood after all!

References:

Bickerstaff, K., Simmons, P., & Pidgeon, N. (2006). Public perceptions of risk, science and governance: main findings of a qualitative study of six risk cases (Technical Report 06-03): Norwich: Centre for Environmental Risk.

Collins, H. M., & Evans, R. (2002). The third wave of science studies: studies of expertise and experience. Social Studies of Science, 32(2), 235-296.

Michael, M. (1992). Lay discourses of science: science-in-general, science-in-particular, and self. Science, Technology & Human Values, 17(3), 313-333.

Partridge, T.et al. 2019. Disturbed earth: Conceptions of the deep underground in shale extraction deliberations in the US and UK. Environmental Values 28(6), pp. 641-663.

Seigo, Selma L'Orange, et al. "Predictors of risk and benefit perception of carbon capture and storage (CCS) in regions with different stages of deployment." International

Journal of Greenhouse Gas Control 25 (2014): 23-32.

Thomas, M.et al. 2017. Public perceptions of hydraulic fracturing for shale gas and oil in the United States and Canada. WIRES Climate Change 8(3), pp. e450. (10.1002/wcc.450)

Wynne, B. (1996). May the Sheep Safely Graze? A Reflexive View of the Expert-Lay Knowledge Divide. In S. Lash, B. Szerszynski & B. Wynne (Eds.), Risk, environment and modernity: towards a new ecology (Vol. 40). London: Sage.
* * *

---

## Referee Comment (RC2) · Anonymous Referee #2 · 27 Jan 2020

This paper presents new data concerning the contrast in perceptions of geoscience between geoscientists and the lay public, highlighting the role of affect in a mental models approach. The results present an interesting view of an important topic, namely the role of identity and emotion in influencing risk communications between experts and non-experts. Though the results of the paper are interesting, I have some questions regarding the nature of the study that I think need answering before publication.

Firstly, the authors present data in response to the stimulus to "sketch the ground beneath your feet" and then "make sketches of drilling, mining/quarrying and flooding" and these results were analyzed collectively, except for the affective component, where

the flooding data was missing. My question is about the inclusion of the flooding data in the analysis at all. Firstly the dataset for flooding is not complete, given the missing affective survey results, and secondly the type of hazard here is very different to those anthropogenic hazards of commercial geoscience. Thus, unless another (more natural) hazard was also included (such as landslides?) as a comparison, it feels like the stimulus would be related to different conceptualizations of risk and that would confuse the final results.

Secondly in the presentation of the affective beliefs of the geoscientists, the authors state that "the geoscientists have more positive affective responses to mining/quarrying", etc and I am curious how much of that was related to their employment within those fields? It has been shown (such as in Mearns and Flin, 1995) that people working in an industry are more likely to operate from within their own specific and subjective risk framework which is often more positive about the risk than the objective assessment would be, particularly as beneficial employment prospects contribute to mitigating the perceived risk. Therefore if those geoscientists surveyed worked in mining and quarrying fields, it is reasonable that their more positive assessment of the activity could equally be related to their employment, which would be useful information in the context of this study.

Overall, I do think this paper has value but I would like these points addressed before publication.

Additional notes: Line 75: open parenthesis Ref from above: Mearns and Flin (1995) Risk perception and attitudes to safety by personnel in the offshore oil and gas industry: a review, Journal of Loss Prevention in the Process Industries, 8, p299-305
* * *

---

## Author Response (AR1)

[revised manuscript text omitted]

**Reply to Editor**

Dear Editor,

Thank you very much for your helpful and thorough review of our manuscript. We have replied to your comments below (with editorial comments in bold and our comments below each one; please note, in our replies we include line numbers referring to the annotated manuscript version above) and have made changes to the manuscript. These changes can be seen both in the annotated version and in a clean copy. Please note that both these manuscripts also incorporate the previous changes we have made after peer review, as well as these latest changes ones after editorial review. We look forward to your feedback and thank you again for your consideration of our work.

Best wishes,

Anthea Lacchia on behalf of all co-authors
* * *
**Response to Editorial Comments**

**General comment:**

**I caution the authors against drawing conclusions beyond the scope of this study. For example, I wonder if the outcomes would be different with, for example, a non-geoscientist population that benefits from oil and gas extraction, or a geoscientist population that is focused more on basic research and less on industry.**

We take the editor's point and do not wish to suggest that our conclusions are valid for all geoscientists or non-geoscientists. We have addressed this by mentioning the make-up of our sample at several points throughout the manuscript (for instance, see lines 18, 160-167, 170-171, 713-727) and making clear the limitations of this study (we have added a section entitled *Limitations* in the discussion at lines 713-727 in annotated manuscript above). Please see comments below for further details in response to this.

**Specific comments (Page/Line):**

**12 Evidence for this statement, and does this communication struggle go both ways as stated?**

We have now provided evidence in the form of examples of studies reporting communication issues between geoscientists and non-geoscientists in the introduction (lines 41-64). As such, we suggest leaving this sentence in the abstract as it is (i.e., line 12 of annotated manuscript: 'Geoscientists and non-geoscientists often struggle to communicate with each other.') as a means of introducing the broad topic of the paper.

**15 delete space before .**

This has been done.

**18 after (n=38), say where your sample set is from, as it is pretty specific**

We have done this now by adding 'recruited in Ireland' in line 18.

**21 Should be edited from "mental models of non-geoscientists focus more on" to "mental models of the non-geoscientists focused more on," as you cannot generalize your findings out to all geoscientists.**

Thank you for pointing this out. We agree and have made this change (lines 21 and 29).

**22 see comment above for (1/21) and change to "the geoscientists focused"**

This has been done.

**23 this human interactions interpretation seems thin to me. Human interactions with… the environment? Or do you mean the role of humans incl. geoscientists…?**

We agree this could be confusing, so in line 19 of the abstract we have changed 'human interactions' to 'presence of humans', which is how we have loosely defined them. Later in the manuscript (lines 476-479), we are more specific in defining human interactions as 'the presence of humans or human-operated machines in the sketches, comments or labels, including human-built structures such as a field, road or house.' This is the definition we found most useful during qualitative thematic analysis of the sketches.

**23 mental models in general, or mental models of geoscientists vs. non-geoscientists?**

**Be careful to keep your conclusions within the scope of what your study actually**

**addressed.**

Thank you, we have added 'of geoscientists and non-geoscientists' to line 30 to clarify this.

**24 "both components need"?**

Yes, we have added 'both components' to line 31.

**37 understanding**

We have changed the phrasing here from 'a starting point to understand each other is to investigate the differences in mental models between geoscientists (defined as anyone with at least a university degree in geology or geoscience) and non-geoscientists (those without such a degree)' to 'A starting point from which to understand each other is to investigate the differences between geoscientists (defined as anyone with at least a university degree in geology or geoscience) and non-geoscientists (those without such a degree)' (line 68).

**38 you use the term mental models before defining it; you should define it on first use.**

We have changed the phrasing (see comment above) to make sure we define it on first mention (a very simple definition is in the abstract, line 16).

**38 defined for our purposes, or which we define for this study as – make it clear that you**

**are the ones defining it**

We have added 'for our purposes' (line 72).

**41 correct the grammar (need a connecting word after the ,)**

Thank you – we have added 'or' as a connecting word (line 72).

**43 is the concept of mental models specific to the geosciences?**

No, quite right. We have clarified that this study (Libarkin et al) looked at mental models 'in the context of science education' (line 75).

**44 I'm not clear on how these models differ; examples would help**

We have added examples to clarify this. Specifically, we have rephrased to: 'Libarkin et al. (2003) recognise four categories of cognitive (mental) models: "conceptual models" which are precise, highly-stable representations of the world used by geoscientists (for instance, aquifer models); "conceptual frameworks", organised and stable models of the world used by geoscientists (for instance, the notion of gravity); "naïve mental models", intuitive models of the world that so-called 'novices' fill with fragmented and unconnected knowledge (for instance, the notion that the Earth is flat); and "unstable mental models", unstable, incomplete and inexact mental models which are used by novices and easily modified (for instance, the idea that the Earth is spherical, but with flattened portions where humans live)' (lines 75-89).

**[54 – 72 have been replaced]**

**73 contribution = goal?**

Yes, we have changed this (line 154: 'the main goal of the present study').

**73-80 this would benefit from acknowledging some of the specifics of this study, acknowledging that this study; a study of this limited scope cannot aspire to investigate this issue for all of geoscientists and non-geoscientists in all settings (and indeed, how mental models vary between different types of geoscientists, different demographics, and communities with different experiences would all be interesting questions to explore)**

Yes, we agree and do not wish to claim to generalise to all geoscientists and non-geoscientists. We have rephrased (lines 164-167): 'While our sample of geoscientists (n=24) working across Ireland and non-geoscientists (n=38) recruited in a rural community in Ireland is not representative of all geoscientists and non-geoscientists in all settings,  we suggest that understanding differences and resemblances of both the cognitive and affective components of mental models of geoscientists and non-geoscientists can help to improve two-way communication between them about often-contested areas of the geosciences.'

**76-80 introducing an argument in your introduction, rather than posing this idea as a**

**hypothesis or question, gives the strong impression that you entered your study with a preconceived / expected outcome**

The rephrasing in the point above solves this issue.

**83-85 again, this needs to acknowledge the specifics of the study**

We have added 'Irish' here in this introductory sentence (line 171), and more details of sample make up are just below (line 173).

**94-95 this is unclear**

We have added details about the design as follows (lines 181-184): 'In our analyses, we used a mixed experimental set-up of between-subjects design (to compare geoscientists vs non-geoscientists) and within-subjects design (to investigate sketches of subsurface, drilling, mining/quarrying, flooding within our sample group of geoscientists or non-geoscientists).'

**97 which beliefs?**

We have clarified that this refers to 'beliefs about the subsurface and geological activities' (lines 194-195).

**97-98 subject-verb agreement**

Thanks, we have fixed this.

**98 is there a reference for the qualitative thematic analysis technique?**

Yes, we have added references here (line 196): '(Boyatzis, 1998; Marshall and Rossman, 1999).'

**99 the IBM**

Thanks, we have fixed this.

**108 speak to range as well as majority**

We have added details on the range: 'The geoscientists who took part in the study ranged in age from 21 to 59' (lines 203-204) and, for non-geoscientists, 'their age ranged from 16 to 60 or Over' (lines 207-208).

**110 Table 1 – why did you include income and household type?**

Thank you for pointing this out. We gathered this as part of our sociodemographic data but did not analyse it or use it in reporting our findings. We have removed it.

**125 name specific university**

We have edited this to say 'University College Dublin' (line 231).

**133 (n=11), and**

We have added 'and' (line 238).

**145 quarrying, and flooding - the Oxford comma will make this much easier to understand (I recommend applying it throughout the manuscript)**

Thank you, we agree. Have made this change throughout.

**147 follow-up**

We have made this change (line 254).

**150 is there a word missing?**

Yes, 'for' was missing. We have added this to line 257.

**152-153 is there a word or phrase missing? Or ranging -> range?**

Yes, 'were measured' was missing. We have added this now to line 259.

**154 including flooding? I believe you have taken flooding out of your current draft; if so, ignore comment**

Indeed, we have deleted flooding out of the current draft and removed the quantitative sketch analysis and explain why. We still include a description of what we have done, including the questions on flooding which yielded significant results, in the Materials and methods section, lines 256-258, where we have added: 'Flooding did not yield reliable scales for affective responses or significant results for perceived impact, hence it was excluded from further analyses and from the rest of the results.'

**159-163 rearrange sentence for clarity**

This has been done and changed to 'They indicated which feeling they identified with.. etc.' in lines 266-267.

**164 (2018), and**

We have made this change (added 'and', line 271).

**165 "formed both reliable scales" – unclear what this means**

This is based on statistical analysis using Cronbach's Alpha. Changed to 'both formed reliable scales (Table 2), which is indicated by scores of Cronbach's Alpha of 0.70 or higher (Peterson, 1994).' – line 272-274, and added the reference to the Reference list.

**169 Mean (M) and Standard Deviation (SD)**

Added this to line 290.

**169 use of title case (or not) should be consistent throughout**

We have fixed this in Table 2.

**169 why not list perceived impact and affective responses in the order in which they**

**appear in the table?**

We have done this.

**170+ reformat table for readability**

We have done this.

**179 authors**

We have made this change (line 301).

**184 remove ,**

We have made this change (line 316).

**186 indicators, Independent**

We have made this change (line 318).

**187 the IBM**

We have made this change (line 319).

**192 variables ?**

Yes, we have fixed this (line 325).

**195 non-geoscientists in regard to…**

We have added that the tests were run to compare 'cognitive and affective responses of geoscientists and non-geoscientists (line 328).

**200 interactions in regards to what?**

We have rephrased to 'human interactions (presence of humans in the sketches)' in order to explain what this means. A more complete definition is at the start of the section on Human Interactions (line 475).

**201 in the sketches only? Or through the sketches and interviews? Here you refer only to the sketches.**

Good point, in the sketches and written comments on the survey (added to line 331).

**217 0.006], and more**

We have taken out the statistical test results from this paragraph (about the ANOVA repeated measures test) because we decided to remove the quantitative analysis of the sketch analysis once we removed flooding.

**219 remove ' as non-geoscientists refers to the people, not their sketches**

As above.

**221 Fig 1a or just Figure 1?**

Good point: 'Figure 1' is correct (line 351).

**224 comments and sketches ?**

Yes, we have changed this (line 406).

**227 from an anthropocentric**

We have made this change (line 409).

**223 remove (Fig. 1b)**

We have made this change.

**240 denoting is probably the wrong word here**

Good point, we have changed to 'indicating' (line 446).

**240-241 among whom, and in what context?**

Upon further consideration, we have decided to remove this sentence ('This sense of unfamiliarity with the subsurface and geological timescales was also noted by Stewart (2016)'), since the study mentioned was more related to sustainable geoscience and not as relevant to our finding about non-geoscientists.

**247 subject-verb agreement**

We have fixed this.

**254 is this reflecting different beliefs or different knowledge? On what do you base the difference in beliefs? It seems that non-geoscientists aren't expressing a difference in beliefs, but rather that beyond a certain point they just don't know. Or are you saying there are different beliefs within both groups?**

This is an interesting point. We agree it is probably more appropriate to use the term knowledge here, incorporating the idea of technical knowledge and lay expertise. Beliefs is a term more appropriate later on in the manuscript when talking about environment and economy, for instance. We have rephrased as follows (lines 456-468): 'Therefore, while at first glance it appears that geoscientists possess conceptual mental models and non-geoscientists possess naïve mental models, given that geoscientists have more familiarity and technical knowledge related to geoscience, we find that underlying this, the mental models of both geoscientists and non-geoscientists are complex and reflect different knowledge in both groups.'

**257 quarrying; and g,h**

We have fixed this and changed this part of the caption (line 474) to 'g,h, subsurface (left) and mining/quarrying (right).'

**264 repeated measures analysis ?**

We have removed this paragraph as it was part of the quantitative sketch analysis.

**264 what does main effect mean? Is this a known social science construct?**

We have removed this paragraph as it was part of the quantitative sketch analysis.

**267 these are human, not geological processes – this is a significant distinction, as humans play an essential role in drilling and mining/quarrying, where they (we) may play no role in geohazards such as flooding. It's important to consider this in the interpretation of the mental models.**

Yes, thank you for pointing this out. Indeed, these are human processes. We recognise that this is important in interpreting our results and have noted this in our interpretation (e.g., lines 490-493 of annotated manuscript: 'A possible explanation is that mining/quarrying and drilling are tied to geoscientists' jobs and therefore including humans in the sketches may be geoscientists' way of highlighting the social process of science and their work.'). We have also changed the word 'processes' to 'activities' throughout the manuscript to better reflect this point.

**283-284 this seems like a stretch. The physical acts of mining/quarrying and drilling are not research endeavors in the same way that basic research is. They are focused on the human process and not on the Earth process (which would be, e.g., sedimentation, compaction, metamorphism, orogeny, etc., not mining or drilling)**

We have removed this sentence.

**286 again, they are human, not geological, processes**

Noted (see our responses above).

**295-300 cannot generalize your findings to all geoscientists; change have to had (x2: 295, 297)**

We have made this change (lines 525 and 527).

**307 negative responses to what?**

We have deleted this paragraph.

**309 that the geoscientists of our study have**

We have deleted this paragraph.

**312 what would geoscientists' affective response have to do with their misperceptions of**

**others? Is there evidence that they do misperceive the affective responses of nongeoscientists?**

Upon further consideration, this is unclear and have deleted this paragraph (i.e. the paragraph: 'Recent research (Perlaviciute et al., 2017) indicates that negative responses from members of the general public are often overrepresented in the media. This, combined with our result that geoscientists have fewer negative affective and more positive affective responses to geological processes like drilling and mining/quarrying than non-geoscientists, explains why geoscientists may misperceive affective responses of non-geoscientists.')

**326 the non-geoscientists**

We have made this change (line 580).

**326-327 "relate their negative emotions with the negative impact of geoscience on the environment" - unclear what this means**

We have changed this sentence to: 'it was clear that the non-geoscientists tended to associate negative emotions with the negative impact of geoscience on the environment.' - lines 579-580.

**357 do you want to indicate gender?**

Thank you for pointing this out, we have removed this as it is not necessary for the purposes of interpreting results and we wish to preserve anonymity (line 626).

**367 tended**

We have made this change.

**368 tended**

We have made this change.

**371 acknowledge your sample set, not necessarily indicative of all geoscientists & nongeoscientists in all circumstances**

We have changed to 'a sample of Irish geoscientists and non-geoscientists' (line 646) to make this clear.

**378 evidence? You have not argued this clearly yet**

We have added this sentence to argue our point: 'Indeed, we find that the mental models of both groups are complex reflections of different knowledge, beliefs and affect.' (line 654-656).

**383-384 move reference after "reality"?**

We have done this (line 660).

**394 this is a relatively strong unsubstantiated statement**

Noted. We have changed to the sentence (line 677) to 'As a practical step, in communicating with each other, geoscientists and non-geoscientists may wish to acknowledge their differences and focus on commonalities in order to find common ground', as well as adding examples in the form of suggestions' (also see below).

**396-397 why, if they contract with those of non-geoscientists? Clarify your reasoning**

Thank you for pointing out this was unclear. We have added the following (lines 679-685) by way of explanation: 'For instance, given that both geoscientists and non-geoscientists are concerned with the impacts of geoscience on the economy and the environment and given that both groups incorporate affect in their mental models of geoscience concepts and activities, geoscientists may be able to reach wider audiences by acknowledging these concerns and affective components, and including feelings and affect in their chosen form of communication (e.g., personal motivations for their research).'

**398 evidence? You bring the communications arguments in without substantial tie-in to the study or references. Consider statements such as "may benefit from" if you would like to make arguments.**

See previous point, we addressed this in the sentences added above.

**401-402 similarly, this is a very strong statement and referred to as a finding, when this is clearly an opinion; you have not addressed this issue in your study in any way, or if you have you have not made it clear through the description of your work**

We have changed to: 'In addition, geoscientists may benefit from using storytelling and narrative, which typically include both affective and cognitive components, as their chosen modes of communication, a recommendation consistent with previous science communication research (Dahlstrom, 2015).' - (lines 685-688).

**403-405 Why? Address this.**

We have rephrased (lines 695-698) to: 'Given that non-geoscientists often incorporate lay expertise in their mental models, in order to build trust and common ground, geoscientists may also wish to acknowledge and tap into local knowledge held by non-geoscientists, for example simply by asking non-geoscientists questions about their local area.'

**409 How about any benefits to geoscientists in recognizing the affective component of their mental models? Might it change how they see themselves as keepers of knowledge?**

Thank you for this very interesting suggestion. We agree that this would be a benefit and have added this paragraph (lines 688-694): 'If geoscientists acknowledge the emotional component of their mental models, this may also lead them to reflect on the meaning of scientific knowledge and to change their view of themselves as keepers of knowledge. On one hand, this could influence how they communicate their work and activities to geoscientists and non-geoscientists, but it could also lead to a broader understanding of epistemology and the social component of geoscience on the part of geoscientists (see Stewart 2016).'

**416 This is not a finding; of course, we are all human! Be more specific about what you mean.**

We have changed our opening paragraph of the conclusions (lines 731-737) to: 'Our finding that geoscientists stray beyond facts into the realm of emotions and beliefs in constructing their mental models of geoscience concepts and activities is a key realisation for geoscience communication practitioners. We have argued that putting the human element at the centre of communication strategies will help achieve meaningful dialogue between geoscientists and non-geoscientists.'

**Additional changes:**

-We have changed the term 'processes' to 'activities' throughout the manuscript, to be more precise since, as rightly pointed out by the Editor, mining/quarrying and drilling are human - activities and not geological processes (such as metamorphism, orogeny, etc).

-We have altered Fig. 1g,h (following the removal of flooding from the image).
* * *
**Reply to Editor and Referees (submitted on February 2020)**

Dear Editor,

Thank you for your time and consideration of our work. We wish to thank the referees for a thorough and helpful analysis of our findings. Below, we have responded to each point made by the referees, indicating proposed changes we wish to make to the manuscript. We have indicated the page and line number of the original manuscript where the proposed changes apply (e.g., (3/23) = page 3, line 23). Referee comments are included in italics in the left column of the table below, followed by our responses and changes in the manuscript in the right column below.

Thank you and best wishes,

Anthea Lacchia on behalf of all authors

**Responses to Referee Comments**
*Response to Anonymous Referee 1*

| Referee comments | Response from authors and changes in the manuscript |
|---|---|
| *"Thank you for the opportunity to review this article, which has the potential to make a useful contribution to the field of geoscience communication. The paper is based on a sound idea and appropriate methods, but it needs work before it will be ready for publication."* | We thank Referee 1 for the positive response to the idea behind this study and its methods. We in turn have found this review very helpful and informative. |

| | |
|---|---|
| *"Much more engagement with the literature around perceptions of geosciences is necessary."* | We agree with Referee 1's suggestion and have made use of the detailed and useful list of references provided both within and at the end of the review, many of which are now included in the manuscript. We have also added further references beyond those suggested. We detail the suggested additions to the manuscript below, grouped according to the suggestions made by Referee 1. |
| *"For example, for expert and lay perceptions of underground geology see Partridge et al (2019); Seigo et al (2014)."* | We read both papers with interest. We include a reference to Partridge et al (2019) in the section on Impact on Economy and Environment, and a discussion of Seigo et al (2014) in the introduction. Specifically, we have altered the introduction to include more literature discussion, as follows (3/31): "However, geoscientists often struggle to communicate with non-geoscientists, particularly around controversial topics such as resource extraction and risk communication. For instance, past studies have investigated public perception and risk communication in the case of fracking (e.g. Boudet *et al*., 2014; Thomas *et al*., 2017), carbon capture and storage (Seigo et al., 2014) and earthquakes (e.g. Marincioni *et al*., 2012). Specifically, in the context of earthquake risk communication, Marincioni *et al.* (2012) studied the case of the 2009 earthquake in l'Aquila, Italy, as a result of which 308 people died: the authors identified a lack of clear communication from the risk management authorities to the public in relation to earthquake prediction and structural resistance of buildings. In the context of public perception of carbon capture and storage, Seigo *et al.* (2014) compared risk and benefit perceptions of the technology in different Canadian regions, and found that predictors of risk perceptions, such as sustainability concerns, did not vary across different regions and were unrelated to familiarity with the technology. The authors also point out that there is a need to address lay people's "misconceptions" related to carbon capture and storage, in order for informed decisions to take place. In the context of a public perceptions of fracking, Thomas *et al*., 2017, in a literature review, identified mixed levels of awareness of shale operations, as well as ethical issues and widespread distrust of responsible |

| | parties. Other studies concerning fracking, such as that by Boudet *et al.* (2014), which looked at public perceptions of fracking in the U.S., found differences in perception between different genders, socioeconomic backgrounds, income levels and level of education, and highlighted a need for "wide ranging and inclusive public dialogue" around the risks and benefits of fracking. |
|---|---|
| | We have added the following to (20/363): |
| | "In line with previous studies of perceptions of the underground (Partridge *et al*., 2019), we recognised tensions between economic values and environmental values in comments written on the survey, such as "*Drilling for a well for water is ok. Drilling for oil or gas is not necessary. Invest in solar and wind energy alternatives. Fracking is just idiotic.*" Such comments tended to equate fracking with a threat, associated with fear. Another participant wrote: "*Concerned about fracking if not properly supervised*". This tension may be linked to a desire for control (cf Hooks *et al*. 2019) and regulation of geoscience activities and technologies (e.g., GSI, 2016), as typified by comments such as "*Concerned about fracking if not properly supervised*" or "*Groundwater pollution with farming practices, I would like it to be more controlled.*" Geoscientists, while indicating an awareness of the negative effects of geoscience on the environment in written comments on the survey, generally downplayed *etc*." [as before] |
| *"The conclusion that mental models are the result of beliefs that include both cognitive and affective components is not new. In two of the papers that you cite for example, the authors describe a number of 'nonknowledge' factors that contribute to risk perceptions – and you need to engage more with this literature (Sjoberg et al, 2007; Thomas et al., 2015)."* | We agree that the papers dealing with emotions and risk perceptions should be highlighted clearly in our paper and propose altering and adding to the introduction to mental models literature from (3/54) to (3/72) as follows: |
| | "Mental models have previously been used to understand non-experts' perceptions of geoscience-related topics. For instance, Bostrom *et al.* (1994) investigated non-experts' mental models of climate change, and found that global warming was regarded as "both bad and highly likely". Zaunbrecher *et al.,* (2018), investigating non-experts' mental models of geothermal energy, identified varying attitudes and knowledge levels among participants, with negative emotions being evoked by the concepts of drilling and |

power stations. These studies also stress that there are emotional or affective components underlying the mental models of non-experts. However, most mental models studies focus merely on cognitive components (e.g. Gibson *et al*., 2016; Goel, 2007; Johnson-Laird, 2010, 2013; Shipton *et al.,* 2019) or on the cognitive superiority of geoscientists over non-geoscientists (Libarkin *et al*., 2003; Vosniadou and Brewer, 1992). Here, we argue that mental models should also incorporate subjective and affective representations of a phenomenon, for both geoscientist and non-geoscientists.

Affect is a general positive or negative feeling that people may experience about an event, a situation, a technology or a process (Finucane *et al.,* 2000). An affective response is thus the response to such an event, situation, technology or process, based on positive or negative feelings. Misperceptions of geological activities among the public are often attributed to affective and emotional processes (Devine-Wright, 2005; Finucane *et al.,* 2000; Loewenstein *et al*., 2001). The role of emotions in risk perception and communication around nuclear waste has been investigated by Sjöberg (2007), who argued that emotions such as interest play an important role in risk perception and attitude. In Zaunbrecher *et al*.'s (2018) study of public perception of geothermal energy, an association between positive emotions and the acceptance of geothermal energy was identified. Similarly, Thomas *et al.* (2015) identified negative emotions in the mental models of non-experts when considering sea level change. While these studies recognise emotions as a component of the mental models of non-geoscientists, far less is known about the affective responses of geoscientists, and how they influence their mental models, as well as how they compare with those of non-geoscientists.

Compared with the number of studies focusing on non-experts or publics, fewer studies have used mental models to compare experts' and non-experts' perceptions. For example, Gibson *et al.* (2016) identified mismatches in perceptions of subsurface hydrology and geohazards between experts and non-experts. In a study comparing experts' and non-experts' mental models of nuclear waste, Skarlatidou *et al*. (2012) described non-experts' negative perceptions of nuclear

| | waste as co-existing with a positive attitude towards nuclear energy, as well as lack of knowledge and familiarity, and discussed implications for risk communication. In the context of sea-level change, Thomas *et al.* (2015) identified both consistencies between the mental models of experts and non-experts, and barriers to publics engaging with the issue, and argued that factors other than knowledge bear an influence on the mental models of non-experts. These factors include "levels of concern, perceptions of self-efficacy and responsibility, trust and ways of actively engaging with or avoiding the issue" (Thomas *et al.*, 2015, p.78). |
|---|---|
| *"The conclusion that experts are 'human' and have affective responses is also not new: see for example Wynne (1996) for a discussion of lay expertise. Some critical engagement with what constitutes expertise would also be helpful – see for example Collins and Evans (2002)."* | We have added a section devoted to lay expertise (which we include in this response to Referee 1 below). |
| *"There are major biases in your sample: the geoscientists are much younger, largely students, predominantly male, and highly educated. How do you know that your results are not a function of these differences rather than the fact that they are geoscientists? A wealth of research shows that risk perceptions vary with age, gender etc – and this should be taken much more into account, as this raises serious questions for your results and conclusions."* | We recognise that these factors relating to our sample could introduce bias and affect our results. The makeup of our sample of geoscientists was also mentioned by Referee 2.

Given these limitations, we agree with the suggestion that it is helpful to focus more on our qualitative findings than we did in the original manuscript. We also suggest adding the following section at the end of our Discussion (22/408):

"*Limitations*

While this mixed-method study highlights differences and similarities between the mental models of geoscientists and non-geoscientists, it should be noted that the sample size is small, and thus our results need to be interpreted with care. Future research is needed to validate our conclusions. It should further be noted that the geoscientists who took part in this study were primarily highly-educated males working in applied geoscience research at the time the survey took place (only 2 worked outside of research), and they were younger compared to the non-geoscientists who took part (for details, |

see Materials and Methods). The latter is fairly representative for geoscientists (e.g., Dutt *et al.*, 2016), however, we cannot say with certainty that these differences in socio-demographics play a role in the differences we find. For example, female and younger geoscientists may hold different perceptions of geoscience activities and their impacts (cf. Seigo *et al.*, 2014). However, this does not influence our main conclusion that geoscientists' mental models are influenced by both cognitive and affective responses."

We also propose adding this sentence to the Method section (5/108):

"We discuss the limitations associated with our sample in the Discussion section."
* * *
*"You have some interesting qualitative findings here that deserve much more discussion. For example, what local knowledge was included? What can this tell us? What is the significance of this? Why did experts include more labels – is this anything to do with fulfilling what was expected of them? Perhaps they enjoyed it more than the lay participants so wanted to provide as much information as possible? Are geoscientists more practised in drawing diagrams, and might this explain the attention to detail? Does the amount of detail in the pictures reflect a lack of understanding or a perceived lack of understanding (the 'I'm not a scientist so I don't know' phenomenon. . . - see for example Bickerstaff et al 2006; Michael, 1992). Is it indifference or ignorance? There are so many things here that I would like you tell me more about. Due to the nature of your sample, I think it is difficult for you to focus on the quantitative results, but you could certainly explore your qualitative results more."*

We thank Referee 1 for their helpful comments. We agree that our qualitative results merit further emphasis. In our revision, we wish to retain our quantitative results relating to human interactions, affective response and impact on economy and environment, as they are a useful exploratory tool, but to focus more on qualitative results. In particular, we propose describing the sketches from a qualitative point of view, which allows us to explore interesting themes such as lay expertise.

(12/203): we wish to change the heading from 'Technical knowledge and familiarity' to 'Knowledge and expertise' so as to better reflect the lay expertise now discussed.
We also suggest adding subsections on 'technical knowledge and familiarity', and on 'lay expertise', and one entitled 'Concluding remarks'.

We include below our new proposed section entitled 'Knowledge and expertise':

[revised manuscript text omitted]

| | |
|---|---|
| | Since we no longer focus on the quantitative results, we wish to move the paragraph starting on (12/208) and ending (12/214) to the method, at (11/180), and delete from (11/182) to (11/184).

We also wish to add 'lay expertise' to this sentence of the abstract to highlight our findings (1/20):

"While the mental models of non-geoscientists focus more on the perceived negative environmental and economic impacts of geoscience, as well as providing evidence of lay expertise, those of geoscientists focus more on human interactions." |
| *(2/29) provide some examples of why geoscience is integral in society e.g. mining, risk management, landscape management, etc. etc.* | We suggest changing (2/29) to:

"Geoscience activities such as mining, quarrying, hazard risk management and landscape management are an integral part of society, affecting local communities, citizens and scientists." |
| *(2/33) provide examples of problems with geoscience communication, such as with fracking and geohazards (e.g. L'Aquila earthquake).* | We have added the example of risk communication related to earthquakes and the example of L'Aquila to the introduction (already mentioned above). |
| *(3/54-55) the term 'expert' would be more appropriate than 'geoscientist' as not all of this research looks at geoscientists.* | We agree and have replaced the terms 'geoscientist' and 'non-geoscientists' with 'expert' and 'non-expert' in this sentence. |
| *(3/65) as the authors do in Thomas et al (2015, cited above).* | Our revised introduction adds this. |
| *(19/334) you mention lack of trust – you could relate this with previous research that also discusses lack of trust in geoscience industry (e.g. Thomas et al., 2017).* | We wish to add this sentence to (19/335): "Lack of trust in industry and government has previously been identified as a dominant theme in a review of public perceptions of hydraulic fracturing for shale gas and oil (Thomas et al., 2017)." |

*Response to Anonymous Referee 2*

| Referee comments | Response from authors and changes in the manuscrips |
|---|---|
| *"This paper presents new data concerning the contrast in perceptions of geoscience between geoscientists and the lay public, highlighting the role of affect in a mental models approach. The results present an interesting view of an important topic, namely the role of identity and emotion in influencing risk communications between experts and non-experts. Though the results of the paper are interesting, I have some questions regarding the nature of the study that I think need answering before publication."* | We thank Referee 2 for these positive comments about our study and for providing a very helpful review. |
| *"Firstly, the authors present data in response to the stimulus to "sketch the ground beneath your feet" and then "make sketches of drilling, mining/quarrying and flooding" and these results were analyzed collectively, except for the affective component, where the flooding data was missing. My question is about the inclusion of the flooding data in the analysis at all. Firstly the dataset for flooding is not complete, given the missing affective survey results, and secondly the type of hazard here is very different to those anthropogenic hazards of commercial geoscience. Thus, unless another (more natural) hazard was also included (such as landslides?) as a comparison, it feels like the stimulus would be related to different conceptualizations of risk and that would confuse the final results."* | Referee 2 is correct in pointing out that flooding was omitted from the affective component analysis. We wish to also point out that it was also omitted from the analysis on impact on the economy and environment. Upon consideration, we agree that flooding, as a hazard, is quite a different category to mining/quarrying and drilling. The comparison with landslides would have indeed been interesting but beyond the scope of this paper. With this in mind, we omitted flooding from the sketch analysis (described at 10/175), when looking for differences between geoscientists and non-geoscientists (based on the indicators number of labels, layers, sense of scale, technical jargon and depth) and re-ran the ANOVA Repeated Measures analysis, but the multivariate tests were not always significant ($p \geq 0.05$) without the flooding data. We thus propose removing the quantitative sketch analysis from our paper, focusing instead on qualitative analysis, and removing flooding from the results altogether so as not to confound results.

**Proposed changes in detail:**

Delete (11/179) to (11/187) since results are to be discussed qualitatively.

Delete 'flooding' from the manuscript in the following lines and pages: 1/17; 9/154; 9/159. |

| | Propose deleting sketches g,h from Fig. 1, and substituting them with examples of a geoscientist's sketch showing stick figures with smiling faces, and a non-geoscientist showing evidence for lay expertise.

Modify 8/149 to: "Flooding did not yield reliable scales for affective responses or significant results for perceived impact, hence it was excluded from further analyses and from the rest of the results."

Delete the following paragraph from 'human interactions', 16/264:
"An ANOVA repeated measures revealed a significant main effect of human interaction across the sketches of drilling, mining/quarrying and flooding, (Wilks' $\lambda$ = 0.51); [$F(2, 53)$ = 25.02, $p \leq 0.001$], and showed more human interactions in the sketches of geological processes (drilling and mining/quarrying) compared to geohazards (flooding), ($p \leq 0.001$). "

Since we re-analysed the data without flooding, we wish to modify the sentence starting with 'interestingly' at (16/268) to the following:

"Interestingly, geoscientists included more human interactions than non-geoscientists when sketching drilling, [$t(56)$ = 3.77, $p \leq 0.001$] and mining/quarrying, [$t(56)$ = 3.14, $p$ = 0.003]." |
|---|---|
| *"Secondly in the presentation of the affective beliefs of the geoscientists, the authors state that "the geoscientists have more positive affective responses to mining/quarrying", etc and I am curious how much of that was related to their employment within those fields? It has been shown (such as in Mearns and Flin, 1995) that people working in an industry are more likely to operate from within* | Our sample of geoscientists was mainly made up of people working in research concerned with applied geoscience such as mining/quarrying, though we did not formally gather this data. Though these people were not directly working in those industries, it is indeed possible that this could have affected their risk perceptions and also their affective responses. Otherwise, the notion that the |

| | |
|---|---|
| *their own specific and subjective risk framework which is often more positive about the risk than the objective assessment would be, particularly as beneficial employment prospects contribute to mitigating the perceived risk. Therefore if those geoscientists surveyed worked in mining and quarrying fields, it is reasonable that their more positive assessment of the activity could equally be related to their employment, which would be useful information in the context of this study."* | profession of geoscientists (and their interest and enjoyment) would affect our results was actually one of our hypotheses: that geoscientists would differ from non-geoscientists due to their profession.

To discuss this further, we propose adding the following to (18/312):

"It should be pointed out that many of the geoscientists in our sample worked in research in geoscience activities (though area of research was not formally gathered), which could have resulted in more positive affective associations with their field of research, such as feelings of safety (cf. Mearns & Flin, 1995)." |
| *"Additional notes: Line 75: open parenthesis"* | We have fixed this typo. |

---

## Referee Report (RR1)

Overall I really enjoyed this paper and it utilised a slightly different approach to its analysis than a traditional mental models paper, many of which use interview data alongside textual/pictorial analyses. The sketches were really interesting with fascinating analysis and they incorporated an affective component of geoscientists, something that is not usually done for m.m approach but provides a worthy discussion point. There are a few clarifications and changes that I feel would improve this paper.

**24** Not clear what 'beliefs' are just that they are composed of cognitive and affective components. A sentence defining how the authors want to contextualise 'beliefs' would be useful somewhere in the introduction

**59 – 61** A couple of lines acknowledging that non-geoscientists may actually be very knowledgeable from a different perspective (as the paper does end up doing later on) should be included. There are clearly those without a degree who will be at an 'expert' level for example. How one defines a non-expert and expert can be quite tricky!

**108** Is there any work that has explored affective responses of experts more generally even? This is something that seems to be lacking in the literature more generally. And is there a reason that research focuses on simply the knowledge aspect of how this group understand a risk issue? (I mean generally, not this specific paper as it is a departure from the norm for m.m)

**158** Switch non-geoscientists and scientists (number of) for consistency as you report recruitment of geoscientists and then non-geoscientists over the page (139-140), just to ease reading

**167** Table 1 does not add up, numbers reported in all sociodemographic categories (age and educational level) for both groups do not come to the total recruited. Please fix.

**176** I was curious why you chose to recruit from an area where you anticipated knowledge levels about the topic to be higher, was this a conscious decision? The aim of the research and purpose for creating the mental models could be clearer. You do mention in your final paragraph of the introduction (124 – 133) that you use a rural community but give the impression that the sample is typical, not one with potentially strong links to this topic. Make this a little clearer ie. that you used this sample specifically because they lived in this area and therefore would be more aware of possible issues or raise things geoscientists may not have considered for example and therefore future dialogue with such a community should consider x, y, z (if this was your intention)

**178** Please include a sentence on what information respondents were given in their recruitment letter, was it topic blind or not? (To make it clear if this would bias response)

**237** The authors state that they pre-defined six indicators, on what basis was this done? Was this prior to the thematic analysis or from the TA? Not clear, particularly as late on (263) they mention there being four themes. Were the initial six themes deductive and latter inductive utilising grounded theory? (e.g. see Glaser and Strauss (1967) or Pidgeon and Henwood (1997))

**342** Check Nature reference, e.g. title?

**485, 503, 517** I would argue that the definition of mental models are conceptual or knowledge based but as you correctly say, of course this is not the way humans think and there is research that supports that particularly in non-experts/publics conceptualisation of risk even where it is unfamiliar to them. This is where your paper is more unique as experts are not traditionally asked about their affective response in this methodology but you have done just that. My main thought throughout this paper has been blurring of this line of whether mental models are used as a tool for experts to

only provide rational information or if they should be incorporating emotion (as you discuss later, you want them to include emotion). I think this discussion point is one that will be divisive. Clearly it has an impact on how an expert conceptualises a risk issue and is an important consideration. You state that perhaps they should provide motivations and affect in their dialogues in communication strategies, this might get tricky when it comes to impartiality or a reliance on expertise. I think this is a worthwhile discussion point and one that is quite difficult. I am thinking of deliberative work where people defer to the experts and the influence of affect and how this would interact. In summary, acknowledge the complexities that this may bring.

**533** Related to the previous point about communications, are geoscientists the group that should be doing this as they may be seen as having an agenda? So what is the purpose of the communications, make this a little clearer. Perhaps a more interdisciplinary approach including social scientists and comms specialists would be appropriate to eradicate the 'expert' on said topic and influence they have.

---

## Editor Decision (ED1)

Editorial comments for "The human side of geoscientists: comparing geoscientists' and non-geoscientists' cognitive and affective responses to geology."

This paper is an interesting presentation of frameworks on earth science concepts held by geoscientists vs. non-geoscientists. While many critical factors were commented on by the two reviewers and addressed by the authors, I, as the editor managing this work, would like to ensure the following issues are addressed before publication. Some or even many of these issues may have been addressed already by the authors, but, lacking a full draft of a rewrite to review, it is difficult to evaluate.

Overall, this work may contribute to effective geoscience communication practices by highlighting factors to consider when communicating geoscience with non-geoscientists. I caution the authors against drawing conclusions beyond the scope of this study. For example, I wonder if the outcomes would be different with, for example, a non-geoscientist population that benefits from oil and gas extraction, or a geoscientist population that is focused more on basic research and less on industry.

Specific comments (Page/Line):
Evidence for this statement, and does this communication struggle go both ways as stated?
delete space before .
after (n=38), say where your sample set is from, as it is pretty specific
Should be edited from "mental models of non-geoscientists focus more on" to "mental models of the non-geoscientists focused more on," as you cannot generalize your findings out to all geoscientists.
see comment above for (1/21) and change to "the geoscientists focused"
this human interactions interpretation seems thin to me. Human interactions with… the environment? Or do you mean the role of humans incl. geoscientists…?
mental models in general, or mental models of geoscientists vs. non-geoscientists? Be careful to keep your conclusions within the scope of what your study actually addressed.
"both components need"?
understanding
you use the term mental models before defining it; you should define it on first use.
defined for our purposes, or which we define for this study as – make it clear that you are the ones defining it
correct the grammar (need a connecting word after the ,)
is the concept of mental models specific to the geosciences?
I'm not clear on how these models differ; examples would help
[54 – 72 have been replaced]
contribution = goal?
73-80   this would benefit from acknowledging some of the specifics of this study, acknowledging that this study; a study of this limited scope cannot aspire to investigate this issue for all of geoscientists and non-geoscientists in all settings (and indeed, how mental models vary between different types of geoscientists, different demographics, and communities with different experiences would all be interesting questions to explore)

76-80   introducing an argument in your introduction, rather than posing this idea as a hypothesis or question, gives the strong impression that you entered your study with a preconceived / expected outcome

83-85   again, this needs to acknowledge the specifics of the study

94-95   this is unclear which beliefs?

97-98   subject-verb agreement is there a reference for the qualitative thematic analysis technique?

the IBM

speak to range as well as majority

Table 1 – why did you include income and household type?

name specific university (n=11), and quarrying, and flooding  - the Oxford comma will make this much easier to understand (I recommend applying it throughout the manuscript)

follow-up is there a word missing?

152-153        is there a word or phrase missing? Or ranging -> range?

including flooding? I believe you have taken flooding out of your current draft; if so, ignore comment

159-163        rearrange sentence for clarity (2018), and

"formed both reliable scales" – unclear what this means

Mean (M) and Standard Deviation (SD)

use of title case (or not) should be consistent throughout why not list perceived impact and affective responses in the order in which they appear in the table?

170+    reformat table for readability authors remove , indicators, Independent the IBM

variables  ?

non-geoscientists in regard to…

interactions in regards to what?

in the sketches only? Or through the sketches and interviews? Here you refer only to the sketches.

0.006], and more remove ' as non-geoscientists refers to the people, not their sketches

Fig 1a or just Figure 1?

comments and sketches  ?

from an anthropocentric remove (Fig. 1b)

denoting is probably the wrong word here

240-241 among whom, and in what context?

subject-verb agreement is this reflecting different *beliefs* or different *knowledge*? On what do you base the difference in beliefs? It seems that non-geoscientists aren't expressing a difference in beliefs, but rather that beyond a certain point they just don't know. Or are you saying there are different beliefs *within* both groups?

quarrying; and g,h repeated measures analysis ?

what does main effect mean? Is this a known social science construct?

these are human, not geological processes – this is a significant distinction, as humans play an essential role in drilling and mining/quarrying, where they (we) may play no role in geohazards such as flooding. It's important to consider this in the interpretation of the mental models.

283-284 this seems like a stretch. The physical acts of mining/quarrying and drilling are not research endeavors in the same way that basic research is. They are focused on the human process and not on the Earth process (which would be, e.g., sedimentation, compaction, metamorphism, orogeny, etc., not mining or drilling)

again, they are human, not geological, processes

295-300 cannot generalize your findings to all geoscientists; change have to had (x2: 295, 297)

negative responses to what?

that the geoscientists of our study have what would geoscientists' affective response have to do with their misperceptions of others? Is there evidence that they do misperceive the affective responses of non-geoscientists?

the non-geoscientists

326-327 "relate their negative emotions with the negative impact of geoscience on the environment" - unclear what this means do you want to indicate gender?

tended tended acknowledge your sample set, not necessarily indicative of all geoscientists & non-geoscientists in all circumstances evidence? You have not argued this clearly yet

383-384 move reference after "reality"?

this is a relatively strong unsubstantiated statement

396-397 why, if they contract with those of non-geoscientists? Clarify your reasoning evidence? You bring the communications arguments in without substantial tie-in to the study or references. Consider statements such as "may benefit from" if you would like to make arguments.

401-402        similarly, this is a very strong statement and referred to as a finding, when this is clearly an opinion; you have not addressed this issue in your study in any way, or if you have you have not made it clear through the description of your work

403-405        Why? Address this.

How about any benefits to geoscientists in recognizing the affective component of their mental models? Might it change how they see themselves as keepers of knowledge?

This is not a finding; of course, we are all human! Be more specific about what you mean.

---

## Author Response (AR2)

[revised manuscript text omitted]

**Author response to Reviewer #3**

Overall I really enjoyed this paper and it utilised a slightly different approach to its analysis than a traditional mental models paper, many of which use interview data alongside textual/pictorial analyses. The sketches were really interesting with fascinating analysis and they incorporated an affective component of geoscientists, something that is not usually done for m.m approach but provides a worthy discussion point. There are a few clarifications and changes that I feel would improve this paper.

We wish to thank the referee for their positive comments and constructive review. We have enjoyed and benefited from reading the reviewer comments and the references therein. We have made all the changes recommended and think the paper has benefited from the review. Please find replies to each comment below in blue.

**24** Not clear what 'beliefs' are just that they are composed of cognitive and affective components. A sentence defining how the authors want to contextualise 'beliefs' would be useful somewhere in the introduction

We have added the following sentence to the introduction (lines 134-135), explaining what we take beliefs to mean (based on a definition that is used in science education):

"We define beliefs as "psychologically-held understandings, premises or propositions about the world that are felt to be true" (Richardson, 1996, p. 103)."

We have also added a reference to the study cited in the reference list.

**59 – 61** A couple of lines acknowledging that non-geoscientists may actually be very knowledgeable from a different perspective (as the paper does end up doing later on)

should be included. There are clearly those without a degree who will be at an 'expert'

level for example. How one defines a nonexpert and expert can be quite tricky!

We completely agree that these definitions are tricky and that those who may not be defined formally as 'experts' may possess expert knowledge (and indeed find this to be the case in our results). We have rephrased as follows (lines 59-65):

"A starting point from which to understand each other is to investigate the differences between geoscientists, defined as anyone with at least a university degree in geology or geoscience, and non-geoscientists, defined as those without such a degree. While acknowledging that those without a degree in geoscience may well possess expert knowledge relating to geoscience, we choose to adopt these definitions as indicators of expertise, and as useful starting points from which to discuss differences and similarities."

**108** Is there any work that has explored affective responses of experts more generally even? This is something that seems to be lacking in the literature more generally. And is there a reason that research focuses on simply the knowledge aspect of how this group understand a risk issue? (I mean generally, not this specific paper as it is a departure from the norm for m.m)

We agree that, while there is plenty of psychological research on risk perception that highlights the importance of both emotions and cognitive components (e.g., Sjöberg

2007, Slovic *et al* 2004), studies exploring affective responses of experts seem to be lacking from the literature. Apart from one study about climate change (Lowe &

Lorenzoni, 2007) which recognises experts' affective components in risk perception, to our knowledge the mental models literature tends to focus on cognitive components when looking at experts. Perhaps this focus on knowledge for experts is related to the highly-pervasive notion that experts are objective and somehow devoid of emotion, which leads researchers (often from the natural sciences) to view other experts as objective (we touch on this in lines 539-540 of the paper). We hope that future work will explore this interesting avenue of research.

Reference mentioned above:

Lowe, T.D. and Lorenzoni, I. Danger is all around: Eliciting expert perceptions for managing climate change through a mental models approach. *Global Environmental*

*Change*, 17, 1. 131-146, 2007.

**158** Switch non-geoscientists and scientists (number of) for consistency as you report recruitment of geoscientists and then non-geoscientists over the page (139-140), just to ease reading

We have changed to '24 geoscientist and 38 non-geoscientist participants' (line 174).

**167** Table 1 does not add up, numbers reported in all sociodemographic categories (age and educational level) for both groups do not come to the total recruited. Please fix.

Thank you for noticing this error. We have fixed this and added a row with the number of respondents who declined to disclose their age (n=1). All counts add up now.

**176** I was curious why you chose to recruit from an area where you anticipated knowledge levels about the topic to be higher, was this a conscious decision? The aim of the research and purpose for creating the mental models could be clearer. You do mention in your final paragraph of the introduction (124 – 133) that you use a rural community but give the impression that the sample is typical, not one with potentially strong links to this topic. Make this a little clearer ie. that you used this sample specifically because they lived in this area and therefore would be more aware of possible issues or raise things geoscientists may not have considered for example and therefore future dialogue with such a community should consider x, y, z (if this was your intention)

This is a good point. We chose this area because it is the current focus of public engagement activities by geoscientists working in iCRAG (Irish Centre for Research in

Applied Geoscience) and we wished to better understand the audience for these activities.

We have added the following to the introduction (137-142):

"We chose to recruit participants from a rural community in Ireland where geologists typically conduct fieldwork (Martinsen *et al.*, 2017) because the area's spectacular

Carboniferous geology lends itself to public engagement events. Better understanding the community will allow geoscientists and public engagement practitioners to develop such public engagement activities."

And rephrased the following sentence in the Materials & Methods (lines 196-199):

"Given the popularity of the area with geologists, we also anticipated that non- geoscientists living in the area may have a relatively high level of familiarity with geology or with groups of geologists, thus potentially providing useful insights for dialogue in this community."

**178** Please include a sentence on what information respondents were given in their recruitment letter, was it topic blind or not? (To make it clear if this would bias response)

We have added (lines 202-205): "In the invitation letters, participants were asked to take part in a study investigating public perception of geology, including knowledge about the geology of Co. Clare and the subsurface. No specific information on the aims of our study was provided in order to minimise response bias."

**237** The authors state that they pre-defined six indicators, on what basis was this done? Was this prior to the thematic analysis or from the TA? Not clear, particularly as late on (263) they mention there being four themes. Were the initial six themes deductive and latter inductive utilising grounded theory? (e.g. see Glaser and Strauss (1967) or Pidgeon and Henwood (1997))

Thank you for pointing out that this was unclear. The pre-defined six indicators were developed first, and the thematic analysis (from which resulted four themes) was conducted afterwards, and informed by the indicator analysis. Both types of analysis were done using an inductive grounded theory approach. We have rephrased as follows to better explain our methodology (lines 260-285):

"The first and second authors examined the sketches using a grounded theory approach taken as "the progressive identification and integration of categories of meaning from data" (Willig, 2008, p.35). This allowed the identification of six indicators of knowledge and familiarity in the sketches, namely: amount of *technical jargon*, defined as the presence of technical and subject-specific vocabulary in the labels of sketches, s*ense of scale,* which refers to an indication of the awareness of the size of different elements included in the sketches (usually provided by a point of reference such as a scale bar); *number of layers,* the number of layers of rock or other material in the sketches; *number*

*of labels,* the number of labels included in the sketches; *depth*, which refers to the depth to which they sketched the subsurface, ranging from the ground surface (coded as 1) to the core (5); and *human interactions*. The authors scored the sketches independently based on this. Pearson's correlation was used to determine the inter-rater reliability, which was deemed acceptable (Pearson's $r \geq 0.7$, $p \leq 0.001$).

To test the differences between geoscientists and non-geoscientists on the six pre- defined indicators, Independent Sample T-tests and ANOVA Repeated Measures analyses were conducted using the IBM SPSS Statistics 24 software package.

These results informed our qualitative analysis of the sketches, whereby the sketches were subsequently analysed by means of thematic analysis to identify themes that were common to some or all of the sketches (Boyatzis, 1998; Marshall and Rossman, 1999).

Thematic analyses were conducted manually by the first author."

We added this reference to the reference list:

Willig, C. Grounded theory. In: Willig, C. (Eds.), *Introducing qualitative research in*

*psychology – adventures in theory and method*. Maidenhead: Open University Press –

McGraw Hill Education, 34-51, 2008.

**342** Check Nature reference, e.g. title?

We have added the title of this *Nature* editorial (The best research is produced when researchers and communities work together) in the references (line 700).

**485, 503, 517** I would argue that the definition of mental models are conceptual or knowledge based but as you correctly say, of course this is not the way humans think and there is research that supports that particularly in non-experts/publics conceptualisation of risk even where it is unfamiliar to them. This is where your paper is more unique as experts are not traditionally asked about their affective response in this methodology but you have done just that. My main thought throughout this paper has been blurring of this line of whether mental models are used as a tool for experts to only provide rational information or if they should be incorporating emotion (as you discuss later, you want them to include emotion). I think this discussion point is one that will be divisive. Clearly it has an impact on how an expert conceptualises a risk issue and is an important consideration. You state that perhaps they should provide motivations and affect in their dialogues in communication strategies, this might get tricky when it comes to impartiality or a reliance on expertise. I think this is a worthwhile discussion point and one that is quite difficult. I am thinking of deliberative work where people defer to the experts and the influence of affect and how this would interact. In summary, acknowledge the complexities that this may bring.

This is a very interesting point and indeed how this will work in practise, whether the communication is taking place in deliberative workshops or dilemma cafés, is complex and merits further research. We have added these lines (lines 564-572) to the discussion:

"While it is useful for geoscientists to acknowledge or reflect on the affective components of their mental models, whether it is always appropriate to incorporate emotions in communication efforts is a complex matter that is likely to depend on the mode of communication (e.g., in person workshops versus an explainer video on social media). There may well be occasions when the purpose of a science communication or public engagement activity is limited to information sharing. We suggest that, in these cases, the self-reflection brought about by the internal acknowledgement of affective components will still be of benefit to the geoscientists engaging in these activities."

**533** Related to the previous point about communications, are geoscientists the group that should be doing this as they may be seen as having an agenda? So what is the purpose of the communications, make this a little clearer. Perhaps a more interdisciplinary approach including social scientists and comms specialists would be appropriate to eradicate the 'expert' on said topic and influence they have.

We completely agree that there is a key role here that can be played by science communication and public engagement specialists. We have added the following sentence (lines 579-583):

"The above recommendations are also very relevant to public engagement and science communication practitioners who not only will be trained in how to engage with communities and publics, but are also less likely to be seen as having an agenda (for instance motivated by economic interests or links to industry) by the non-geoscientists they are engaging with."